# **Insights Into Mesoscale Eddy Dynamics: A Three-Dimensional Perspective on Potential Density Anomalies**

Yan Barabinot<sup>1</sup>, Sabrina Speich<sup>1</sup>, and Xavier Carton<sup>2</sup>

<sup>1</sup>Ecole Normale Supérieure, Laboratoire de Météorologie Dynamique (LMD), IPSL, ENS-PSL, 24 rue Lhomond, Paris 75005, France

<sup>2</sup>Université de Bretagne Occidentale (UBO), Laboratoire d'Océanographie Physique et Spatiale (LOPS), IUEM, rue Dumont Durville, Plouzané 29280, France

**Correspondence:** Yan Barabinot (yan.barabinot@lmd.ipsl.fr)

Abstract. Mesoscale eddies are fundamental components of the global ocean circulation. *In situ* observations and Lagrangian analyses have shown that some eddies are materially coherent, transporting within their cores a water mass distinct from the surrounding environment. Additionally, laboratory experiments indicate that eddies locally deflect isopycnal surfaces in accordance with thermal wind balance, regardless of whether they trap a water mass. These two mechanisms drive *spiciness mode anomalies* and *heaving mode anomalies*, respectively, associated with mesoscale eddies. In this study, we quantitatively assess the physical processes governing mesoscale eddy dynamics by introducing a novel theoretical decomposition of the potential density field within eddy cores that accounts for both effects. We apply this framework to six anticyclonic eddies sampled during the EUREC4A-OA, METEOR 124, and Physindien 2011 oceanographic campaigns. Unlike previous studies, we evaluate not only the amplitude of these anomalies but also their vertical structure. Our results confirm that *heaving mode anomalies* dominate the total density anomaly. However, contrary to previous assumptions, we demonstrate that their vertical structure is dictated by the local background stratification and often exhibits a nearly Gaussian profile. In contrast, *spiciness anomalies* provide only a second-order contribution to the total potential density anomaly, making them negligible for most dynamical processes governing mesoscale eddies. By bridging experimental results with observational eddy datasets, our study refines the understanding of the mesoscale eddy vertical structure, providing a more accurate predictive framework for their shape and role in the transport of oceanic properties.

## 1 Introduction

Ocean dynamics is inherently nonlinear, involving a wide range of physical processes that generate dynamical structures spanning spatial scales from 1 km to over 1,000 km and beyond. Among these, mesoscale eddies —coherent vortices with spatial scales of 10–200 km and lifetimes of 10–100 days or more (Carton, 2001; Chelton et al., 2011; Morrow and Traon, 2012)—play a fundamental role in ocean circulation and transport. Numerical simulations and laboratory experiments have demonstrated that eddies in a rotating stratified fluid exhibit significantly longer lifetimes than those in a non-rotating homogeneous environment (McWilliams, 1984, 1989, 1990; Aubert et al., 2012).

Two key mechanisms contribute to their persistence. First, stratification constrains the velocity field horizontally, and the predominantly two-dimensional nature of the rotating flow leads to an upscale energy cascade, favoring the formation of large eddies (Kolmogorov, 1941; McWilliams, 1984). Second, in a stratified ocean, eddy flow is governed by thermal wind balance (Douglass and Richman, 2015; Cao et al., 2023; Penven et al., 2014), where local modifications in stratification induce radial buoyancy gradients that balance vertical velocity gradients. This balance holds when the Rossby number remains below unity, ensuring that buoyancy anomalies persist and sustain the rotating flow. These buoyancy-driven anomalies, commonly referred to as *heaving mode anomalies* (Bindoff and Mcdougall, 1994; Durack and Wijffels, 2010; Häkkinen et al., 2016; Lv et al., 2023), arise due to the isopycnal lateral transport of volume anomalies by mesoscale eddies. This process does not alter water properties (such as temperature and salinity) along isopycnal surfaces but instead redistributes density volumes vertically. In an anticyclonic eddy, the core contains an excess volume of water at a given density level compared to the surrounding. This leads to a characteristic bipolar displacement of isopycnals. In the upper part of the eddy, isopycnals are displaced upward as lighter water accumulates. In the lower part of the eddy, isopycnals are displaced downward due to an increased volume of heavier water at depth. This results in an expansion of density layers across a greater depth range than in the surrounding ocean. The inverse is true for cyclonic eddies that induce a deficit of volume at a given density level.

When flow trajectories form closed loops, lateral intrusions are minimal, and a significant volume of water remains trapped within the eddy core. This trapped water mass retains the thermohaline properties of its formation region and can be transported over long distances (Flierl, 1981; Beron-Vera et al., 2013; Haller, 2015). Recent studies show that, on isopycnal surfaces, both anticyclonic and cyclonic eddies can exhibit positive or negative temperature and salinity anomalies (Aguedjou et al., 2021; Cui et al., 2021; Lin et al., 2019). However, the impact of these anomalies on eddy dynamics remains poorly understood, despite their occasional use as initial conditions in numerical simulations (de Marez et al., 2020). These anomalies, termed *spiciness anomalies* (Bindoff and Mcdougall, 1994; Durack and Wijffels, 2010; Häkkinen et al., 2016; Lv et al., 2023), were first introduced by Stommel (1962); Munk (1981) to describe thermohaline variations on isopycnal surfaces (Jackett and McDougall, 1985; Flament, 2002; Huang, 2011). Spiciness anomalies represent actual heat and salt (and other properties) transport by eddies, whereas heaving mode anomalies correspond to isothermal and isohaline displacements.

Conversely, some experimental studies have reported eddies that lack thermohaline anomalies on isopycnal surfaces. Instead, these eddies appear to be associated with vertical displacements of isopycnals relative to a quiescent state (Bonnier et al., 2000; Beckers et al., 2001; Negretti and Billant, 2013; Mahdinia et al., 2017). Such thermohaline anomalies arise when there is a mismatch between the water mass in the eddy core and its surroundings due to either eddy displacement (Carton et al., 2010; L'Hégaret et al., 2016; Barabinot et al., 2024, 2025) or changes in ambient thermohaline properties while the eddy remains stationary (e.g., the Lofoten eddy (Bosse et al., 2019)). These scenarios are difficult to replicate in laboratory experiments.

Although previous studies have distinguished *heaving mode anomalies* from *spiciness mode anomalies* and recognized that spiciness anomalies are generally weaker, their influence on density and eddy dynamics remains an open question. This knowledge gap is crucial for reconstructing velocity fields, assessing eddy stability, delineating flow paths, and quantifying eddy transport and coherence. Our primary objective is to quantitatively characterize density anomalies in mesoscale eddies from a hydrological perspective, rather than a purely dynamical one. This approach is motivated by two key considerations: (i)

potential density, as a scalar field, is more readily measurable with existing oceanographic instruments compared to velocity fields, and (ii) our methodology can be directly applied to Argo float data, facilitating a more detailed three-dimensional reconstruction of eddy structures, particularly their vertical extent.

To address these challenges, we propose a novel theoretical decomposition of potential density that explicitly accounts for the contributions of *heaving mode anomalies* and *spiciness mode anomalies*. Additionally, we develop analytical models for the vertical structure of density anomalies based on quasi-geostrophic and tracer diffusion principles. Building on previous studies (Flierl, 1987; Zhang et al., 2013; Carton et al., 2010; Carton and McWilliams, 1989), we propose analytical functions to describe the universal shape of potential density anomalies, drawing inspiration from experimental results (Bonnier et al., 2000; Beckers et al., 2001; Negretti and Billant, 2013; Mahdinia et al., 2017). Such functions are essential for several applications: (i) initializing or refining numerical simulations, (ii) estimating eddy volumes, and (iii) improving the parameterization of unresolved structures in coarse-resolution ocean models.

Finally, this study bridges the gap between laboratory experiments and real-world mesoscale eddies, aiming to establish a universal formulation applicable to eddies in stratified fluid with varying background stratification. To achieve this, we conduct a detailed analysis of six anticyclonic eddies sampled during the EUREC4A-OA (Stevens et al., 2021; Speich and Team, 2021; L'Hégaret et al., 2022), METEOR 124 (Karstensen and Wölfl, 2016; Karstensen et al., 2016b; Karstensen and Krahmann, 2016), and Physindien 2011 (L'Hégaret and Carton, 2011; L'Hégaret et al., 2016) oceanographic cruises. While our study focuses exclusively on anticyclonic eddies due to data limitations, the theoretical framework is equally applicable to cyclonic eddies.

## 2 Theoretical Framework

#### 2.1 Potential density field decomposition

We consider an isolated, materially coherent vortex in a hydrostatically balanced ocean with no fronts, incropping, or outcropping. A cylindrical coordinate system  $(r,\theta,z)$  is used, where r is the radial coordinate,  $\theta$  the azimuthal angle, and z the vertical coordinate. The conservative temperature and absolute salinity fields are denoted as  $\widetilde{T}(r,\theta,z)$  and  $\widetilde{S}(r,\theta,z)$ , respectively, with the associated potential density  $\widetilde{\sigma}(r,\theta,z)$  at atmospheric pressure. The reference profiles, representing the climatological state of the ocean, are given by  $\widetilde{T}(z)$ ,  $\widetilde{S}(z)$ , and  $\widetilde{\sigma}(z)$ . The stratification is assumed to be stable for invertibility. These reference quantities vary only with depth. For clarity, we distinguish function definitions (e.g.,  $\widetilde{\sigma}$ ) from their evaluated values (e.g.,  $\sigma = \widetilde{\sigma}(r,\theta,z)$ ). The potential density deviation is referenced against pure water density (1000 kg.m<sup>-3</sup>). Under these assumptions,  $\widetilde{\sigma}$  and  $\widetilde{\overline{\sigma}}$  are bijective in z and invertible, allowing us to define the reciprocal functions  $\widetilde{Z}$  and  $\widetilde{\overline{Z}}$  that provide the depth z corresponding to a given potential density  $\sigma$ , such that:

$$z = \widetilde{Z}(r, \theta, \widetilde{\sigma}(r, \theta, z)), \quad z = \widetilde{\overline{Z}}(\widetilde{\overline{\sigma}}(z)).$$
 (1)

Since two water masses of equal density can have different temperature and salinity, we define isopycnal thermohaline anomalies to distinguish trapped water from surrounding water:

90 
$$\forall \sigma, \quad \widetilde{\Delta_{\sigma}T}(r,\theta,\sigma) = \widetilde{T}(r,\theta,\sigma) - \widetilde{\overline{T}}(\sigma),$$
 (2)

$$\forall \sigma, \quad \widetilde{\Delta_{\sigma}S}(r,\theta,\sigma) = \widetilde{S}(r,\theta,\sigma) - \widetilde{\overline{S}}(\sigma). \tag{3}$$

These anomalies quantify temperature and salinity variations along isopycnals, representing the so-called *spiciness mode anomalies* (Bindoff and Mcdougall, 1994; Durack and Wijffels, 2010; Häkkinen et al., 2016; Lv et al., 2023), commonly used to estimate eddy-induced heat and salt transport (Laxenaire et al., 2019, 2020; Yang et al., 2021).

Next, we define modified temperature and salinity fields:

$$\forall z, \quad \widehat{\widetilde{T}}(r,\theta,z) = \widetilde{T}(r,\theta,z) - \widetilde{\Delta_{\sigma}T}(r,\theta,\widetilde{Z}(r,\theta,\widetilde{\sigma}(z))), \tag{4}$$

$$\forall z, \quad \widehat{\widehat{S}}(r,\theta,z) = \widetilde{S}(r,\theta,z) - \widetilde{\Delta_{\sigma}S}(r,\theta,\widetilde{Z}(r,\theta,\widetilde{\sigma}(z))). \tag{5}$$

These fields share the same isopleths as the isopycnal temperature and salinity fields. Since each isopycnal now corresponds to a unique isotherm and isohaline, these fields quantify the so-called *heaving mode anomalies* (Bindoff and Mcdougall, 1994; Durack and Wijffels, 2010; Häkkinen et al., 2016; Lv et al., 2023).

Given a temperature-salinity field, the potential density  $\sigma$  at atmospheric pressure is determined by the nonlinear function  $\widetilde{F}$ :

$$\widetilde{\sigma}(r,\theta,z) = \widetilde{F}(\widetilde{T}(r,\theta,z),\widetilde{S}(r,\theta,z)).$$
 (6)

This function is computed using the Thermodynamic Equation of Seawater - 2010 (McDougall et al., 2003; Roquet et al., 2015). Introducing  $\tilde{T}$  and  $\tilde{S}$  in a Taylor expansion, we write:

$$\widetilde{\sigma} = \widetilde{\widehat{\sigma}} + \delta^2 \widetilde{\sigma},\tag{7}$$

where:

100

$$\widetilde{\widetilde{\sigma}} = \widetilde{F}(\widetilde{\widehat{T}}, \widetilde{\widehat{S}}),$$
 (8)

$$\delta^{2}\widetilde{\sigma} = \left(\frac{\partial \widetilde{F}}{\partial T}\right)_{S,z} \widetilde{\Delta_{\sigma}T} + \left(\frac{\partial \widetilde{F}}{\partial S}\right)_{T,z} \widetilde{\Delta_{\sigma}S} + o(\widetilde{\Delta_{\sigma}T}) + o(\widetilde{\Delta_{\sigma}S}). \tag{9}$$

Physically,  $\delta^2 \widetilde{\sigma}$  represents the density anomaly due to *spiciness mode anomalies*. This term is computed as the difference between  $\widetilde{\sigma}$  and  $\widetilde{\widehat{\sigma}}$  at geopotential levels.

By further subtracting the background stratification  $\tilde{\overline{\sigma}}$ , we define the first-order density anomaly:

$$\forall z \quad \delta \widetilde{\sigma}(r, \theta, z) = \widetilde{\widetilde{\sigma}}(r, \theta, z) - \widetilde{\overline{\sigma}}(z). \tag{10}$$

This term represents the isopycnal displacement due to *heaving mode anomalies*. In mesoscale eddies, these anomalies manifest as shallowing or deepening of isopycnals within the core. Thus, the total potential density  $\tilde{\sigma}$  is decomposed into three components (see Figure 1):

$$\widetilde{\sigma}(r,\theta,z) = \widetilde{\overline{\sigma}}(z) + \delta \widetilde{\sigma}(r,\theta,z) + \delta^2 \widetilde{\sigma}(r,\theta,z). \tag{11}$$

Figure 1. Schematic representation of the potential density field decomposition for surface (a) and subsurface (b) anticyclonic eddies.  $\overline{\sigma}$  represents the background stratification (dark lines indicate isopycnal surfaces),  $\delta \sigma$  corresponds to the *heaving mode anomaly*, and  $\delta^2 \sigma$  to the *spiciness mode anomaly*. Together, they form the total *in situ* field  $\sigma$ .

## 2.2 Density Anomalies Formulation

## 2.2.1 Expression for $\delta \widetilde{\sigma}$

For a pure *heaving mode anomaly*, in cylindrical coordinates,  $\widetilde{\eta}(r,\theta,\sigma)$  is the vertical displacement of isopycnal  $\sigma$  with respect to the mean state. Initially, an isopycnal of value  $\sigma_0$  satisfies:  $\sigma_0 = \widetilde{\overline{\sigma}}(\widetilde{\overline{Z}}(\sigma_0))$ .

After displacement, we construct  $\widetilde{Z}(r,\theta,\sigma_0) = \widetilde{\overline{Z}}(\sigma_0) + \widetilde{\eta}(r,\theta,\sigma_0)$ , but also  $\sigma_0 = \widetilde{\sigma}(\widetilde{Z}(r,\theta,\sigma_0))$ . As we simply moved the isopycnal of value  $\sigma_0$  from its state of rest, we write:

$$\sigma_0 = \widetilde{\sigma}(\widetilde{Z}(r, \theta, \sigma_0)) = \widetilde{\overline{\sigma}}(\widetilde{\overline{Z}}(\sigma_0)), \tag{12}$$

As this expression holds whatever the value of  $\sigma_0$ , we thus have  $\widetilde{\sigma}(\widetilde{Z}) = \widetilde{\overline{\sigma}}(\widetilde{Z} - \widetilde{\eta})$ .

Developing the first order, this leads to

$$\widetilde{\sigma}(\widetilde{Z}) = \widetilde{\overline{\sigma}}(\widetilde{Z}) - \widetilde{\eta} \frac{d\widetilde{\overline{\sigma}}}{d\widetilde{Z}}.$$
(13)

As  $\widetilde{Z}$  gives the geopotential level of every isopycnal, we can finally write:

$$\delta \widetilde{\sigma}(r, \theta, z) = -\widetilde{\eta}(r, \theta, z) \frac{d\widetilde{\overline{\sigma}}}{dz}(z). \tag{14}$$

We have obtained the relationship presented in Bretherton (1966) for potential density anomalies resulting from purely heaving mode. The background stratification influences both the amplitude and vertical structure of  $\delta \tilde{\sigma}$ . Introducing h as the characteristic vertical displacement scale  $\tilde{\eta}$ , we derive the scaling relation:

$$\delta\sigma \approx h \frac{\Delta \overline{\sigma}}{\Delta z}.\tag{15}$$

The vertical gradient of  $\tilde{\overline{\sigma}}$  depends on the oceanic region and whether the eddy is surface- or subsurface-intensified.

For a surface-intensified eddy, we estimate a typical variation of  $1 \text{ kg.m}^{-3}$  over 100 m in the undisturbed state. A vertical displacement of 100 m then yields an anomaly of approximately  $1 \text{ kg.m}^{-3}$  for  $\delta \tilde{\sigma}$ . For a subsurface eddy, such as the one depicted in Figure 4, we assume a background density variation of  $0.5 \text{ kg.m}^{-3}$  over 200 m. Given a characteristic displacement of 200 m, the resulting density anomaly is approximately  $0.5 \text{ kg.m}^{-3}$ .

These estimates illustrate how the stratification profile governs the magnitude of  $\delta \tilde{\sigma}$  and how vertical displacement influences the density anomaly in different oceanic regimes.

# **2.2.2** Expression of $\delta^2 \widetilde{\sigma}$

We now consider a pure *spiciness mode anomaly*. Starting with the definition  $\rho = \sigma + 1000 \text{ kg.m}^{-3}$ , we retain the notation for functions. The background density field takes the form:

$$\widetilde{\overline{\rho}}(z) = \rho_0 \left( 1 - \beta_T (\widetilde{\overline{T}}(z) - T_0) + \beta_S (\widetilde{\overline{S}}(z) - S_0) \right), \tag{16}$$

where  $\rho_0$ ,  $T_0$ , and  $S_0$  are characteristic values of density, temperature, and salinity, respectively, obtained as averages over the studied region. The thermal expansion coefficient is  $\beta_T = O(2 \times 10^{-4}) \text{ K}^{-1}$ , and the haline contraction coefficient is  $\beta_S = O(7.6 \times 10^{-4}) \text{ ppt}^{-1}$ , where ppt denotes parts per thousand.

Introducing  $\sigma^* = \rho_0 - 1000 \text{ kg.m}^{-3}$  and defining  $\alpha_1 = \rho_0 \beta_T$  and  $\alpha_2 = \rho_0 \beta_S$ , the reference potential density profile can be expressed as:

$$\widetilde{\overline{\sigma}}(z) = \sigma^* - \alpha_1(\widetilde{\overline{T}}(z) - T_0) + \alpha_2(\widetilde{\overline{S}}(z) - S_0).$$
 (17)

Similarly, within the eddy core, the local density field in cylindrical coordinates follows:

$$\widetilde{\sigma}(r,\theta,z) = \sigma^* - \alpha_1(\widetilde{T}(r,\theta,z) - T_0) + \alpha_2(\widetilde{S}(r,\theta,z) - S_0). \tag{18}$$

Following an isopycnal of value  $\sigma_0$ , we thus obtain:

$$\sigma_0 = \sigma^* - \alpha_1(\widetilde{\overline{T}}(\sigma_0) - T_0) + \alpha_2(\widetilde{\overline{S}}(\sigma_0) - S_0)$$
(19)

$$\sigma_0 = \sigma^* - \alpha_1(\widetilde{T}(r, \theta, \sigma_0) - T_0) + \alpha_2(\widetilde{S}(r, \theta, \sigma_0) - S_0).$$
 (20)

Combining both equations, we obtain

$$\alpha_1(\widetilde{T}(r,\theta,\sigma_0) - \widetilde{\overline{T}}(\sigma_0)) = \alpha_2(\widetilde{S}(r,\theta,\sigma_0) - \widetilde{\overline{S}}(\sigma_0)) \tag{21}$$

which confirms that temperature and salinity anomalies are compensated along isopycnals (McDougall, 1987).

In terms of geopotential levels, the density anomaly due to a distinct trapped water column is given by:

$$\delta^2 \widetilde{\sigma}(r, \theta, z) = -\alpha_1(\widetilde{T}(r, \theta, z) - \widetilde{\overline{T}}(z)) + \alpha_2(\widetilde{S}(r, \theta, z) - \widetilde{\overline{S}}(z)).$$
 (22)

The reference profiles  $\widetilde{T}(z)$  and  $\widetilde{S}(z)$  strongly influence the vertical structure and magnitude of  $\delta^2 \widetilde{\sigma}$ . For typical anomalies of approximately -1 °C in temperature and -0.2 g.kg<sup>-1</sup> in salinity (see Figure 4), we obtain  $\delta^2 \sigma = 0.048$  kg.m<sup>-3</sup>, which is significantly smaller than  $\delta \sigma$ . This suggests that the difference in water properties between the eddy core and its surroundings has a relatively minor impact on potential density anomalies. The following section will further assess the relative magnitude of these terms using *in situ* observations.

## 2.3 Predicting the Shape of $\delta \tilde{\sigma}$ and $\delta^2 \tilde{\sigma}$

Beyond analyzing the potential density anomalies, this study seeks to characterize the spatial structure of these anomalies within the eddy core. To formulate expressions for  $\delta \tilde{\sigma}$  and  $\delta^2 \tilde{\sigma}$ , we separate variables such that

$$\delta \widetilde{\sigma}(r, \theta, z) = \delta_h \widetilde{\sigma}(r, \theta) \delta_z \widetilde{\sigma}(z), \tag{23}$$

$$170 \quad \delta^2 \widetilde{\sigma}(r, \theta, z) = \delta_h^2 \widetilde{\sigma}(r, \theta) \delta_z^2 \widetilde{\sigma}(z), \tag{24}$$

where subscripts h and z denote the horizontal and vertical components, respectively. In the following, we omit the notation for simplicity.

#### 2.3.1 Horizontal Extent

The horizontal structure of mesoscale eddy streamfunctions is often represented as a sum of azimuthal modes (Gent and McWilliams, 1986; Carton, 2001; de Marez et al., 2020). Accordingly, we decompose  $\delta_h \sigma$  and  $\delta_h^2 \sigma$  as:

$$\delta_h \sigma(r,\theta) = \phi(r) \left( 1 + \sum_{n=1}^{+\infty} \pi_n(r) \cos(n\theta + \varpi_n) \right), \tag{25}$$

$$\delta_h^2 \sigma(r, \theta) = \chi(r) \left( 1 + \sum_{n=1}^{+\infty} \varepsilon_n(r) \cos(n\theta + \epsilon_n) \right), \tag{26}$$

where  $\phi(r)$  and  $\chi(r)$  represent the axisymmetric component, while  $\pi_n(r)$  and  $\varepsilon_n(r)$  are the amplitudes of higher-order azimuthal modes with corresponding phase shifts  $\varpi_n$  and  $\epsilon_n$ .

Analyses of satellite altimetry data indicate that mesoscale eddies are predominantly axisymmetric, as shown in studies such as Chelton et al. (2007); Chaigneau et al. (2009). The leading mode (n = 0) generally dominates, while the n = 1 mode often introduces north-south asymmetry due to the β-effect (Carton, 2001). The n = 2 mode describes elliptical structures, which are more susceptible to baroclinic and barotropic instabilities, potentially leading to tripole formation (Carton and McWilliams, 1989; de Marez et al., 2020). According to Chen et al. (2019), this configuration is one of the most frequently observed in global oceanic datasets.

Concerning  $\phi(r)$ , several studies (Carton and McWilliams, 1989; Gallaire and Chomaz, 2003; Ayouche et al., 2021; Buckingham et al., 2021; Bennani et al., 2022; Barabinot et al., 2024) have proposed an empirical formulation. It has been demonstrated that:

$$\phi(r) = \phi_0 \exp\left(-\left(\frac{r}{R_1}\right)^{\alpha_1}\right),\tag{27}$$

where  $\phi_0$  is the amplitude,  $R_1$  is the radius of maximum velocity, and  $\alpha_1$  is an exponent that varies between 2 and 3 during the eddy's lifetime (Bennani et al., 2022; Ayouche et al., 2021). The case  $\alpha_1=2$  corresponds to the well-known Gaussian vortex, which is self-similar and often associated with turbulent diffusion-dominated dynamics. We could have tested the formula proposed by Zhang et al. (2013), who proposed that  $\phi(r)=(1-(r/R_1)^2)\exp(-(r/R_1)^2)$ . However, the formula introduces a change in the sign of  $\phi$  when  $r\approx 1.8R_1$  that we cannot verify because our *in situ* measurements do not capture the far field of the density anomaly (in the radial direction). Furthermore, this formula does not model steeper  $\phi$  profiles, i.e. when  $\alpha>2$ . A similar expression can be proposed for  $\chi(r)$ :

$$\chi(r) = \chi_0 \exp\left(-\left(\frac{r}{R_2}\right)^{\alpha_2}\right),\tag{28}$$

where  $\chi_0$ ,  $R_2$ , and  $\alpha_2$  define the shape of the density anomaly field associated with *spiciness mode anomalies*.

## 2.3.2 Vertical Extent

## 200 Quasi-Geostrophic Arguments

Theoretical developments presented thus far suggest that  $\delta^2\sigma$  is likely negligible compared to  $\delta\sigma$ . This assumption will be validated through empirical analysis. Consequently,  $\delta\sigma$  is expected to be the primary contributor to the pressure anomaly through hydrostatic equilibrium and quasi-geostrophic (QG) flow. For the QG framework to be applicable, two primary conditions must be satisfied by the flow under consideration. First, the Rossby number,  $Ro = V/(f_0R)$ , where V is the maximum velocity, R its location and  $f_0$  the Coriolis parameter, must be small. Second, the Burger number,  $Bu = (N_0H_1/(f_0R))^2$ , where  $N_0^2$  is the squared buoyancy frequency (stratification), and H represents the amplitude of isopycnal deviation, should be of order one. For a typical mesoscale eddy at mid-latitude, with  $V=1~{\rm m.s^{-1}}$ ,  $f_0=10^{-4}~{\rm s^{-1}}$ , and  $R=100~{\rm km}$ , the Rossby number evaluates to Ro=0.1. The buoyancy frequency,  $N_0$  typically ranges from  $10^{-1}~{\rm s^{-1}}$  to  $10^{-2}~{\rm s^{-1}}$ , and with  $H_1=100~{\rm m}$ , the Burger number Bu ranges between 0.01 and 1. Therefore, the requirement that Bu is of order one may not always be satisfied in practice.

$$\frac{\partial p'}{\partial z} = -g \,\delta\sigma,\tag{29}$$

$$\delta\sigma = -\frac{\rho_0 f_0}{q} \frac{\partial \Psi}{\partial z},\tag{30}$$

where p' is the pressure anomaly and  $\Psi$  is the QG streamfunction. The quasi-geostrophic potential vorticity q follows:

$$q = \nabla_h^2 \Psi + \partial_z \left( \frac{f_0^2}{\overline{N}^2(z)} \partial_z \Psi \right), \tag{31}$$

where  $\overline{N}$  the Brunt-Väisälä frequency of the ocean at rest and  $\nabla_h^2$  is the horizontal Laplacian.

For a circular eddy, by separating variables, we can write,

$$q(r,z,t) = A(t)q_0(r)\Psi_v(z), \tag{32}$$

$$\Psi(r,z,t) = A(t)\Psi_h(r)\Psi_v(z), \tag{33}$$

where A(t),  $q_0(r)$ , and  $\Psi_v(z)$  are unknown functions of time, radial distance, and depth, respectively, and  $\Psi_h(r)$  is an unknown radial function.

Assuming  $\Psi_v(z)$  is an eigenfunction of the QG stratification operator (Flierl, 1987; Carton, 2001; de La Lama et al., 2016) with eigenvalue  $\lambda^2$ , we obtain:

$$\partial_z \left( \frac{f_0^2}{\overline{N}^2(z)} \partial_z \Psi_v \right) + \lambda^2 \Psi_v = 0. \tag{34}$$

This equation is assumed valid even for non-circular eddies. This expression is often solved numerically (de La Lama et al., 2016; Meneghello et al., 2021), while here we search for analytical formulae.

A challenge arises when addressing the nonlinearity with respect to the vertical coordinate. Flierl (1987) proposed a solution of the form:

$$\Psi_v(z) = \sqrt{\frac{\overline{N}}{f_0}} \,\psi(\zeta(z)),\tag{35}$$

where  $\psi$  is an unknown function of the stretched coordinate:

$$\zeta(z) = \int_{-\infty}^{z} \frac{\overline{N}}{f_0} dz,$$
 (36)

which accounts for local stratification and Coriolis effects. Substituting into Equation 34, we obtain:

$$\frac{d^2\psi}{d\zeta^2} + (\lambda^2 + \nu^2(z))\psi(\zeta) = 0, \tag{37}$$

$$\nu^{2}(z) = -\frac{f_{0}^{2}}{2\overline{N}^{3}} \left( \frac{5}{2} \frac{\overline{N}^{2}}{\overline{N}} - \overline{N}^{"} \right), \tag{38}$$

where  $\overline{N}'$  and  $\overline{N}''$  are the first and second derivatives of  $\overline{N}$  with respect to z. To derive 38, we used the relation

$$\partial_z \Psi_v = (\overline{N}'/(2\sqrt{\overline{N}f_0}))\psi(\zeta) + (\overline{N}/f_0)^{3/2}\psi'(\zeta).$$

According to Carton (2001),  $\lambda^2$  scales as  $1/R^2 \approx 10^{-10} \text{ m}^{-2}$  (with R=100 km the eddy radius). Using typical oceanic values of  $N_0=10^{-2} \text{ s}^{-1}$ ,  $H_1=100 \text{ m}$ , and  $f_0=10^{-4} \text{ s}^{-1}$ , we estimate  $\nu^2$  to be on the order of  $10^{-10} \text{ m}^{-2}$ , which is comparable to  $\lambda^2$ . Thus,  $\nu^2$  cannot be neglected.

Defining  $\mu = \sqrt{\lambda^2 + \nu^2}$  as a constant (Zhang et al., 2013), the solution to Equation 37 takes the form:

$$\Psi_v(z) = \Psi_{v0} \sqrt{\frac{\overline{N}}{f_0}} \cos(\mu \zeta(z) + \zeta_0),$$
 (39)

where  $\Psi_{v0}$  and  $\zeta_0$  are constants.

Zhang et al. (2013) further simplified this formulation by omitting the  $\sqrt{\overline{N}}$  term, proposing:

$$\Psi_v(z) = \Psi_{v0} \cos(\mu \zeta(z) + \zeta_0). \tag{40}$$

Finally, using Equation 30, we define the vertical structure of  $\delta_z \sigma$  as:

$$\delta_z \sigma(r, \theta, z) = \frac{\partial \Psi_v(z)}{\partial z}$$
. (41)

## Arguments based on experimental studies

While the QG framework has been widely applied in the literature (Charney and Flierl, 1981; Flierl, 1987), we seek simpler expressions based on diffusion arguments and experimental results. As previously established, the potential density anomaly follows  $\delta \sigma = -\eta(r, \theta, z)(d\overline{\sigma}/dz)$ .

To separate the variables, we define:  $\eta = \delta_h \sigma(r, \theta) \eta_z(z)$  where  $\delta_h \sigma$  is the previously defined horizontal component, and  $\eta_z$  represents the vertical variation of  $\eta$ .

Laboratory experiments conducted in constant background stratification have identified a Gaussian shape for the total  $\delta\sigma$  (Flór, 1994; Beckers et al., 2001; Bonnier et al., 2000; Negretti and Billant, 2013; Mahdinia et al., 2017). A similar structure has been observed in Meddies, which are found at depths where background stratification is nearly uniform (Paillet et al., 1999, 2002; Armi et al., 1989; Carton et al., 2002). Diffusion tends to smooth anomalies in a self-similar manner. Although the present study considers a non-uniform stratification, our formulation should be consistent with these past observations when the stratification is approximately constant.

For an anticyclonic eddy,  $\eta_z$  must be positive in the shallower part of the eddy (above the eddy mid-plane) and negative in the deeper part (below the mid-plane). To satisfy this constraint while ensuring a self-similar structure, we propose:

$$\eta_z(z) = \eta_{z0} \left( \frac{z - z_1}{H_1} \right) \exp\left( -\frac{(z - z_1)^2}{H_1^2} \right),$$
 (42)

where  $z_1$  is the geopotential level of the eddy mid-plane,  $H_1$  is a characteristic depth scale, and  $\eta_{z0}$  is the amplitude.

The formulations given in Equations 39 and 40 are derived from the streamfunction, which integrates the potential density anomaly. This integration can obscure details of the vertical structure, particularly in the case of stacked mesoscale eddies (Laxenaire et al., 2019; Napolitano et al., 2024). Thus, explicitly analyzing the isopycnal deviation is crucial.

For  $\delta_z^2 \sigma$ , unlike  $\eta_z$ , there is no reason for the anomaly to change sign in the vertical direction if the trapped water mass is homogeneous. The water inside the eddy core behaves as a single patch, and turbulent diffusion influences the shape of this anomaly. Following Paillet et al. (2002); Assassi et al. (2016); Ciani et al. (2015), who used Gaussian models, we propose:

$$\delta_z^2 \sigma(z) = \xi_0 \exp\left(-\frac{(z-z_2)^2}{H_2^2}\right),$$
(43)

where  $z_2$  is the depth of the maximum anomaly,  $H_2$  is a characteristic vertical scale, and  $\xi_0$  is the amplitude.

## 270 **2.3.3** Summary

Table 1 summarizes the key equations used in this study to model potential density anomalies associated with *heaving* and *spiciness* mode anomalies.

**Table 1.** Proposed formulas for modeling potential density anomalies due to *heaving* and *spiciness* modes. Gent1986 refers to the model proposed by Gent and McWilliams (1986), Flierl1987 and Flierl1987\_cst refer to the model proposed by Flierl (1987) for respectively a varying and constant stratification, Zhang2013 refers to the model proposed by Zhang et al. (2013) and Bara2024 refers to the models proposed in the present study.

| Mode anomaly | Extent     | Formula                                                                                                                                                                                                             | Name           |
|--------------|------------|---------------------------------------------------------------------------------------------------------------------------------------------------------------------------------------------------------------------|----------------|
| Heaving      | Horizontal | $\delta_h \sigma(r, \theta) = \phi(r) \left( 1 + \sum_{n=1}^{+\infty} \pi_n(r) \cos(n\theta + \varpi_n) \right) $ (44)                                                                                              | Gent1986       |
|              | Vertical   | $\delta_z \sigma(z) = \frac{\rho_0 \Psi_{v0}}{g} \left( \mu \overline{N} \sqrt{\frac{\overline{N}}{f_0}} \sin(\mu \zeta(z) + \zeta_0) - \frac{\overline{N}'}{2\sqrt{N}} \cos(\mu \zeta(z) + \zeta_0) \right) $ (45) | Flierl1987     |
|              | Vertical   | $\delta_z \sigma(z) = \Psi_{v0} \frac{\rho_0 \lambda \dot{N}_0}{g} \sin(\lambda \frac{N_0}{f_0} z + \zeta_0) \tag{46}$                                                                                              | Flierl1987_cst |
|              | Vertical   | $\delta_z \sigma(z) = \Psi_{v0} \frac{\rho_0 \vec{\mu}}{g}  \overline{N} \sin(\mu \zeta(z) + \zeta_0) \tag{47}$                                                                                                     | Zhang2013      |
|              |            | $\delta_z \sigma(z) = -\eta_{z0} \frac{d\overline{\sigma}}{dz} \left(\frac{z - z_1}{H_1}\right) \exp\left(-\frac{(z - z_1)^2}{H_1^2}\right) (48)$                                                                   | Bara2024       |
| Spiciness    | Horizontal | $\delta_h^2 \sigma(r, \theta) = \chi(r) \left( 1 + \sum_{n=1}^{+\infty} \varepsilon_n(r) \cos(n\theta + \epsilon_n) \right) (49)$                                                                                   | Bara2024       |
|              | Vertical   | $\delta_z^2 \sigma(z) = \xi_0 \exp\left(-\frac{(z-z_2)^2}{H_2^2}\right) (50)$                                                                                                                                       | Bara2024       |

## 3 In situ Data and Methods

## 3.1 Collection of in situ Data

The data analyzed in this study were collected during three oceanographic cruises conducted in distinct regions of the world ocean: the EUREC4A-OA campaign along the northern coast of South America, in the Atlantic Ocean, the FS METEOR M124 expedition in the South Atlantic, and the Physindien 2011 experiment in the Arabian Sea. The objective was to obtain a representative dataset of mesoscale eddies at different life cycle stages and in varying oceanic environments to investigate their vertical extent. Each campaign provided hydrological and velocity measurements with sufficient spatial coverage (both horizontally and vertically) to enable a detailed characterization of eddy structure.

In this study, structures with a velocity maximum located above (below) the pycnocline are classified as surface (subsurface) eddies. Three of the six analyzed eddies were sampled using two vertical sections rather than a single section. Figures 2, 3, and Section 4 focus on hydrological and velocity sections that cross the eddy as close to the center as possible. Additional sections are discussed in Section 5.2 to examine axisymmetry and horizontal extent.

## 285 3.1.1 EUREC4A-OA Campaign

The EUREC4A-OA field campaign took place between January 20 and February 20, 2020 (Stevens et al., 2021; Speich and Team, 2021) aboard the French RV *L'Atalante*. This study focuses on two anticyclonic eddies (ACs) sampled along the continental slope of Guyane. One is a surface-intensified NBC ring (Subirade et al., 2023) with a velocity field extending to -150 m. The other is a subsurface-intensified intra-thermocline eddy (Barabinot et al., 2024), with its core located between -200 and -600 m. The latter was sampled using two vertical sections.

Hydrographic observations were obtained using a Conductivity Temperature Depth (CTD) profiler, an underway CTD (uCTD) system, and a Lower Acoustic Doppler Current Profiler (L-ADCP). Additionally, a Moving Vessel Profiler (MVP) provided limited surface-intensified eddy observations (Speich and Team, 2021; L'Hégaret et al., 2022). A total of 25 CT-D/uCTD profiles were used to sample the NBC ring, while 24 profiles were used for the subsurface eddy, with 17 additional profiles from the second section. Velocity measurements were collected using two ship-mounted ADCPs (S-ADCPs) operating at 75 kHz and 38 kHz. Measurement accuracies were:

- CTD:  $\pm 0.002$  °C (temperature) and  $\pm 0.005$  g.kg<sup>-1</sup> (salinity).
- uCTD:  $\pm 0.01$  °C (temperature) and  $\pm 0.02$  g.kg<sup>-1</sup> (salinity).
- S-ADCP:  $\pm 3 \text{ cm.s}^{-1}$  (horizontal velocity).

Further details on the data collection methodology can be found in L'Hégaret et al. (2022).

Sections were composed of vertical profiles sampled at varying distances. The resolution of each section is defined as the mean spacing between successive soundings. For the subsurface AC, hydrographic data (CTD/uCTD) had a horizontal (vertical) resolution of  $8.4~\mathrm{km}$  (1 m), while velocity data from the  $38~\mathrm{kHz}$  S-ADCP had a resolution of  $0.3~\mathrm{km}$  (8 m). For the NBC ring, the hydrographic resolution was  $10.3~\mathrm{km}$  (1 m), and the velocity resolution remained  $0.3~\mathrm{km}$  (8 m). The appropriate resolution is selected in subsequent analyses depending on the property of interest.

## 3.1.2 RV METEOR M124 Expedition

The RV *Meteor* M124 cruise was conducted between February 29 and March 18, 2016 (Karstensen et al., 2016a), crossing the South Atlantic from Cape Town to Rio de Janeiro. This study examines three anticyclonic eddies, identified as Agulhas rings, sampled near the west coast of South Africa. Two of these eddies were sampled with two vertical sections (eddies (e) and (f) in Figure 3). Each eddy was associated with an extremum of absolute dynamic topography from satellite altimetry (Karstensen et al., 2016a).

The vertical structure of these eddies was investigated using uCTD and S-ADCP measurements. Hydrographic properties were obtained from 12, 11, and 11 uCTD profiles for the three eddies, with additional sections consisting of 11 and 13 profiles for eddies (e) and (f), respectively. The hydrographic resolution was 21 km (1 m), while velocity data had a horizontal (vertical) resolution of 0.3 km (32 m).

**Table 2.** Grid resolution of interpolated data and cutoff periods for the three cruises.

| Cruise     | $\Delta x$ [km] | $\Delta z$ [m] | $L_x$ [km] | $L_z$ [m] |
|------------|-----------------|----------------|------------|-----------|
| EUREC4A-OA | 1               | 0.5            | 10         | 10        |
| M124       | 1               | 1              | 25         | 40        |
| Phy11      | 1               | 1              | 10         | 10        |

## 3.1.3 Physindien 2011 Experiment

The Physindien 2011 experiment took place in March 2011, focusing on a surface-intensified AC in the Arabian Sea, near the east coast of Oman. Such eddies typically extend to depths exceeding 300 m. Hydrographic properties were measured using approximately 90 uCTD profiles. The resolution of the hydrographic data was 2 km (1 m), while the velocity data had a horizontal (vertical) resolution of 0.3 km (16 m).

## 3.1.4 Eddy Sampling Considerations

Ensuring that *in situ* sections traverse the eddy centers is critical for minimizing boundary effects and obtaining a representative eddy structure. Figure 2 demonstrates that the data align with the findings of Nencioli et al. (2008), who applied an S-ADCP/L-ADCP eddy center detection method. For eddies sampled with two sections, Figure 2 presents the section closest to the center.

## 325 3.2 Data Processing

To minimize spurious effects, only linear interpolations were applied in the radial x and vertical z directions. The data were then smoothed using a fourth-order low-pass filter (scipy.signal.filt in Python) to ensure a consistent representation of mesoscale features. The selection of the filter threshold is subjective and depends on the characteristic scales of interest. Since this study focuses on mesoscale dynamics, submesoscale processes are neglected, and the filtering thresholds are set to approximately  $10 \, \mathrm{km}$  for the horizontal scale  $(L_x)$  and  $10 \, \mathrm{m}$  for the vertical scale  $(L_z)$ . The cutoff scale is chosen to be larger than the spatial sampling interval to preserve relevant mesoscale structures.

The grid resolution for interpolated data  $(\Delta x, \Delta z)$  and the corresponding cutoff scales  $(L_x \text{ and } L_z)$  for each cruise are presented in table 2. Figure 3 shows vertical cross-sections of the eddy core potential density, corresponding to the velocity sections shown in Figure 2, following the application of the low-pass filter. The second sections of eddies (a), (e) and (f) will be used in the discussion section.

#### 3.3 Climatological Averages

The selection of reference profiles for potential density  $\overline{\sigma}(z)$ , temperature  $\overline{T}(z)$ , and salinity  $\overline{S}(z)$  is a critical step in our decomposition method. This study follows the approach developed by Laxenaire et al. (2019, 2020) for anticyclonic eddies.

Figure 2. Velocity vector field at: -50 m for the surface AC of EUREC4A-OA (a), -150 m for the first AC of M124 (b), -50 m for the surface AC of Physindien 2011 (c), -150 m for the second AC of M124 (d), -300 m for the subsurface AC of EUREC4A-OA (e), -200 m for the third AC of M124 (f). The regional bathymetry from the ETOPO2 dataset (Smith and Sandwell, 1997) is shown in the background as colored shading. The estimated eddy center (yellow square) is determined from observed velocities using the Nencioli et al. (2008) method. Colored contours indicate constant tangential velocity. The center is defined as the point where the mean radial velocity is minimum.

To construct the climatological mean profiles, temperature, salinity, and potential density are averaged over geopotential levels within a domain surrounding the sampled eddy. The eddy center is first identified using the methodology of Nencioli et al. (2008). A  $0.5^{\circ} \times 0.5^{\circ}$  square is then defined around the estimated center, ensuring that the center lies at the intersection of the diagonals.

Within this region, temperature, salinity, and potential density profiles from Argo profiling floats are extracted from the Coriolis.eu.org database. These profiles are averaged over a 20-year period for the corresponding month to account for seasonal variability.

**Figure 3.** Vertical sections of the eddy core potential density. The vertical axis represents depth [m] while the horizontal axis represents distance [km]. The black contours indicate isopycnal surfaces, spaced according to their anticyclonic behavior. (a) Subsurface intensified AC from EUREC4A-OA data. (b) Surface intensified AC from EUREC4A-OA data. (c) Surface intensified AC from Physindien 2011 data. (d), (e) and (f) subsurface ACs sampled during the M124 cruise.

## 3.4 Methodology for Potential Density Field Decomposition on in situ Data

For a two-dimensional vertical section passing through the eddy center, with the horizontal axis denoted as  $\mathbf{x}$  and the vertical axis as  $\mathbf{z}$  (as in *in situ* data collected from ships), the potential density decomposition follows:

$$\sigma(x,z) = \overline{\sigma}(z) + \delta\sigma(x,z) + \delta^2\sigma(x,z). \tag{51}$$

To compute  $\delta\sigma$  and  $\delta^2\sigma$ , a series of steps is performed in a specific order. First, the reference profiles of potential density  $\overline{\sigma}(z)$ , temperature  $\overline{T}(z)$ , and salinity  $\overline{S}(z)$  are determined using the methodology described in the previous section, incorporating Argo float data. The thermohaline anomalies  $\Delta_{\sigma}T$  and  $\Delta_{\sigma}S$  are then computed on isopycnal surfaces using Equations 2 and 3. The values of  $\hat{T}(x,z)$  and  $\hat{S}(x,z)$  are subsequently obtained via interpolation at geopotential levels, as defined in Equations 4 and 5.

The next step involves computing  $\hat{\sigma}(x,z)$  using the TEOS-10 equation of state for seawater. The *in situ* potential density  $\sigma(x,z)$  is then subtracted from  $\hat{\sigma}(x,z)$  at geopotential levels, yielding  $\delta^2\sigma(x,z)$  in accordance with Equation 7. The value of  $\delta\sigma(x,z)$  is then obtained by subtracting  $\overline{\sigma}(z)$  from  $\hat{\sigma}(x,z)$  at geopotential levels, as described in Equation 10. Once  $\delta\sigma(x,z)$  is

determined, it can either be divided by  $d\overline{\sigma}/dz$  to obtain the isopycnal displacement  $\eta_z$  or integrated to obtain the streamfunction  $\Psi_v$ , which is analyzed as a function of the stretched coordinate  $\zeta(z)$ .

Since the available data consists of two-dimensional vertical sections, direct analysis of azimuthal modes is not possible. However, the dominant circular mode can be extracted, and residual modes can be estimated. For eddies (a), (e), and (f) in Figure 3, a second section is used to assess axisymmetry and validate the shape of the circular mode. The next section focuses on the horizontal circular mode, while axisymmetry is discussed in the later discussion section.

Following the separation of variables, the density decomposition can be rewritten as:

$$\sigma(x,z) = \overline{\sigma}(z) - \phi(x)\delta_z\sigma(z) + \chi(x)\delta_z^2\sigma(z), \tag{52}$$

where the horizontal functions  $\phi(x)$  and  $\chi(x)$  are given by:

$$\phi(x) = \exp\left(-\left|\frac{x-x_1}{R_1}\right|^{\alpha_1}\right),\tag{53}$$

$$\chi(x) = \exp\left(-\left|\frac{x - x_2}{R_2}\right|^{\alpha_2}\right),\tag{54}$$

with  $x_1$  and  $x_2$  representing the locations of the maxima  $\phi$  and  $\chi$ , respectively, along the ship's course. For brevity, this function is referred to as the "alpha exponential". The vertical components  $\delta_z \sigma$  and  $\delta_z^2 \sigma$  are given by Equations 45, 46, 47, 48, and 50.

During the analysis, both  $\eta_z$  and  $\delta_z^2 \sigma$  exhibited an offset, indicating a shift by a constant. To account for this, the following adjustments are introduced:

$$\eta_z(z) = \eta_{z0} \left( \frac{z - z_1}{H_1} \right) \exp\left( -\frac{(z - z_1)^2}{H_1^2} \right) + B,$$
(55)

$$\delta_z^2 \sigma(z) = \xi_0 \exp\left(-\frac{(z-z_2)^2}{H_2^2}\right) + D.$$
 (56)

Here, B and D represent small correction terms that do not alter the shape of the anomalies.

For each eddy, these generic expressions are fitted to the data using nonlinear least squares optimization (scipy.optimize.curve\_fin Python), minimizing the root mean square (RMS) error. The parameters estimated include:

- $z_1, H_1, \eta_{z0} \text{ for } \eta_z(z),$
- $z_2, H_2, \xi_0 \text{ for } \delta_z^2 \sigma(z),$
- $x_1, R_1, \alpha_1 \text{ for } \phi(x),$
  - $x_2$ ,  $R_2$ ,  $\alpha_2$  for  $\chi(x)$ ,
  - B, D for offset corrections,
  - $\Psi_{v0}$ ,  $\mu$ , and  $\zeta_0$  for sinusoidal components.

The optimization is performed at the vortex center for both  $\eta_z$  and  $\delta_z^2 \sigma$ , as isopycnal displacements are maximal at this location (where  $\delta \sigma$  is also maximal). Here, the vortex center is the appearing eddy center on a two-dimensional cross-section.

This is the location where the orthogonal velocity to the ship transect is zero, not the eddy center determined using the Nencioli et al. (2008) routine. In case the location of the eddy center varies with depth, we took the average location. For  $\phi$  and  $\chi$ , the optimization is performed at the depths where the amplitudes of  $\eta_z$  and  $\delta_z^2 \sigma$  are maximal. Consequently, vertical optimizations are conducted prior to horizontal ones.

Finally, to assess the accuracy of the proposed expressions, the RMS (Root Mean Square) difference between the data and the theoretical predictions is computed.

## 4 Results

## 4.1 First Case Study: Potential Density Decomposition for the Subsurface AC Sampled During EUREC4A-OA

In this section, we apply the potential density decomposition methodology to the subsurface AC sampled during the EUREC4A-OA experiment. While this approach has been applied to all analyzed eddies, we focus on two representative cases for clarity: the subsurface-intensified AC from EUREC4A-OA (this section) and the surface-intensified AC from Physindien 2011 (next section).

The smoothed *in situ* temperature and salinity fields are shown in Figure 4, panels (a) and (d), along with the computed potential density  $\sigma$  and isopycnals (dark lines). The separation of isopycnals, particularly those at  $26.5 \,\mathrm{kg.m^{-3}}$  and  $27 \,\mathrm{kg.m^{-3}}$ , indicates the anticyclonic nature of the eddy.

The thermohaline anomalies on isopycnals are calculated and interpolated to geopotential levels, as shown in Figure 4, panels (c) and (f). The eddy core exhibits a distinct negative anomaly in temperature and salinity, indicative of a trapped water mass that is colder and less saline than its surroundings. The temperature and salinity anomalies reach approximately -1.6 °C and -0.3 g.kg<sup>-1</sup>, respectively. These anomalies are most pronounced at the eddy center and decrease rapidly at its periphery.

The removal of these anomalies from the total *in situ* thermohaline fields yealds the  $(\hat{T}/\hat{S})$  fields associated with *heaving mode anomalies* as shown in Figure 4 (b) and (e). The associated potential density  $\hat{\sigma}$  is also computed, with its isopycnals resembling those in the  $\sigma$  field. However, small deviations beyond the of 26.5 kg.m<sup>-3</sup> and 27 kg.m<sup>-3</sup> isopycnals highlight the impact of *spiciness mode anomalies* on density. Once these anomalies are removed, the eddy core appears warmer and more saline, consistent with their influence.

Figure 5 depicts the potential density decomposition for the EUREC4A-OA eddy. Panel (a) shows the total potential density field computed from the *in situ* T/S fields, while panel (b) shows the modified potential density field derived from the  $\hat{T}/\hat{S}$  fields. The *heaving mode anomaly*,  $\delta\sigma$ , representing isopycnal displacements in the absence of trapped water, is shown in panel (c). The *spiciness mode anomaly*,  $\delta^2\sigma$ , illustrating the thermohaline anomaly effect, is displayed in panel (d).

The  $\sigma$  and  $\hat{\sigma}$  fields exhibit strong similarities, with comparable amplitudes and isopycnal displacements. The *spiciness mode* anomaly  $\delta^2 \sigma$ , shown in panel (d), has small values, reaching only  $-0.015 \text{ kg.m}^{-3}$  in the eddy core, which is four times smaller than the *heaving mode anomaly*  $\delta \sigma$ , which reaches approximately  $0.5 \text{ kg.m}^{-3}$  in panel (c). This result aligns with theoretical expectations.

Figure 4. Two-dimensional vertical sections through the subsurface eddy core from EUREC4A-OA data. Isopycnals are shown as dark lines. (a) Smoothed *in situ* temperature field. (b) Temperature field without *spiciness mode anomalies* (c) Isopycnal thermal anomaly interpolated to geopotential levels (i.e., *spiciness mode anomaly*). (d), (e) and (f) show analogous fields for salinity. Isopycnals in panels (b) and (e) are plotted using the  $\hat{\sigma}_0$  density field.

The  $\delta\sigma$  field exhibits a sign change at approximately 250 m depth, consistent with the theoretical expectation that isopycnals in an anticyclonic eddy are shallower in the upper core and deeper in the lower core relative to the surrounding water. The modest amplitude of  $\delta^2\sigma$  is attributed to the incomplete compensation of temperature and salinity anomalies on isopycnals. At depths below 300 m, the modulus of  $\delta^2\sigma$  is greater within the eddy core than in the surrounding water, whereas above this depth, the modulus is smaller, as evident from the color contrast in panel (d).

## 4.2 Second Case Study: Potential Density Decomposition for the Surface AC Sampled during Physindien 2011

The results for the surface-intensified AC sampled during the Physindien 2011 experiment are presented in Figure 6, with panel configurations similar to those in Figure 4.

The eddy exhibits a positive *spiciness mode anomaly* in both temperature and salinity, indicating a trapped water mass that is warmer and saltier than its surroundings. The anomalies reach values of approximately  $0.6 \,^{\circ}$ C and  $0.1 \, \mathrm{g.kg^{-1}}$ , respectively. The vertical section reveals a secondary positive maximum for both thermal and haline *spiciness mode anomalies* at  $-250 \, \mathrm{m}$ , while between  $-70 \, \mathrm{m}$  and  $-250 \, \mathrm{m}$ , both anomalies decrease to a minimum at approximately  $-150 \, \mathrm{m}$ . The eddy is surrounded

Figure 5. Potential density decomposition in the eddy core from EUREC4A-OA data. The axes are the same as in Fig. 4. (a) Total potential density field from the in situ T/S fields. (b) Modified potential density field from the  $\hat{T}/\hat{S}$  fields. (c) Heaving mode anomaly from Equation 10. (d) Spiciness mode anomaly from Equation 9.

by a ring of colder and fresher water, as evidenced by the negative anomaly at approximately 400 and 550 km in panels (c) and (f).

Figure 7 presents the potential density decomposition for this eddy. The  $\delta^2 \sigma$  anomaly is larger than that of the EUREC4A-OA eddy but exhibits weaker fluctuations

Panels (a) and (b) of Figure 7 illustrate the similarity between  $\sigma$  and  $\hat{\sigma}_0$ , although minor discrepancies are evident. Notably, differences appear between depths of -50 m and -100 m, coinciding with the location of the warm water. Panel (c) shows that the density anomaly associated with the isopycnal displacement,  $\delta \sigma$ , reaches approximately 0.4 kg.m<sup>-3</sup>, while panel (d) depicts a relatively constant density anomaly of approximately -0.15 kg.m<sup>-3</sup>. Compared to the EUREC4A-OA eddy,  $\delta^2 \sigma$  is larger for this anticyclonic eddy. However, its variations are less pronounced, indicating a smoother spatial distribution of the *spiciness mode anomaly* within the eddy core.

Figure 6. Two-dimensional vertical sections across the eddy core from the 2011 Physindien dataset. Isopycnals are shown as dark contours. (a) Smoothed in situ temperature field. (b) Modified in situ temperature  $\hat{T}$  field, excluding the spiciness mode anomaly. (c) Spiciness mode anomaly in temperature interpolated to geopotential levels. (d), (e) and (f) show analogous fields for salinity. Isopycnals in panels (b) and (e) are plotted using the  $\hat{\sigma}_0$  density field.

## 440 4.3 Shape of the Potential Density Anomalies in the Eddy Core

## 4.3.1 Horizontal Extent of $\delta \sigma$ and $\delta^2 \sigma$

Following the decomposition of the potential density field for each eddy, the horizontal structure of  $\delta\sigma$  and  $\delta^2\sigma$  is analyzed. The results are presented in Figure 8. The parameters  $x_1$ ,  $R_1$ ,  $x_2$ , and  $R_2$  are summarized in Table A1 in Appendix A. The values are converted into a dimensionless form and plotted as a function of the reduced coordinates:  $(x-x_1)/R_1$  for  $\phi$  and  $(x-x_2)/R_2$  for  $\chi$ . The theoretical prediction using an alpha-exponential function is in agreement with the observed profiles of both  $\phi$  and  $\chi$  across all cases, with a root-mean-square deviation of less than 10% of the maximum amplitude, which is unity in this case.

Radially, the density anomaly remains relatively uniform in the eddy core but decreases sharply at the periphery. A large value of  $\alpha_2$  corresponds to a strong horizontal gradient in  $\delta^2\sigma$ . For example, in panel (b),  $\chi$  remains nearly constant (equal to one) within the eddy core and decreases sharply outward, following an  $\alpha_2$ -exponential decay with  $\alpha_2 \sim 4$ . Similarly, in panel (c),  $\phi$  remains close to 1 for x between 60 and 200 km, but decreases rapidly outside of this region, with  $\alpha_2 = 6$ .

Figure 7. Potential density decomposition in the eddy core from Physindien 2011 data. The axes are the same as in Fig.6. (a) Total potential density field from the  $\hat{T}/\hat{S}$  fields. (c) *Heaving mode anomaly* from Equation 10. (d) *Spiciness mode anomaly* from Equation 9.

Notably, the horizontal shapes of  $\delta\sigma$  and  $\delta^2\sigma$  are strikingly similar. The parameters  $R_2$ ,  $x_2$  and  $\alpha_2$  closely match those of  $R_1$ ,  $x_1$  and  $\alpha_1$ , which were previously identified as optimal (on average) for modeling  $\phi$ . For instance, the subsurface AC from EUREC4A-OA has values of  $R_1=87.8$  km,  $x_1=99.6$  km, and  $R_2=85.2$  km,  $x_2=98.4$  km. Given the low RMS errors, the alpha-exponential function serves as a robust predictor of the horizontal variation of  $\delta\sigma$  and  $\delta^2\sigma$ .

## 4.3.2 Vertical extent of $\delta^2 \sigma$

Following the analysis of the horizontal structure, we now examine the vertical extent of  $\delta^2 \sigma$ . A Gaussian shape is proposed to model this distribution, with the results presented in Figure 9.

In each panel, the anomaly exhibits a well-defined extremum, decreasing exponentially with depth. However, for the ACs sampled during M124 and the surface AC sampled during EUREC4A-OA, the Gaussian model only partially captures the structure near the surface. This results in an asymmetrical signal and, in some instances, RMS deviations exceeding 10%. The corresponding parameter values are listed in Appendix A, Table A2. Despite these discrepancies near the surface, the Gaussian model remains a suitable representation of  $(\mathcal{E})$ , as indicated by the overall RMS values. A possible explanation for

Figure 8. Comparison between data and theoretical predictions for  $\phi(x)$  (panels (d), (e) and (f)) and  $\chi(x)$  (panels (a), (b), (c)). Each panel shows a horizontal transect in [km] with the dashed red curve representing the alpha-exponential model. Colors correspond to different anticyclonic eddies, as specified in the panel captions. Curves are grouped by  $\alpha_1$  and  $\alpha_2$  values.

Figure 9. Comparison between the *in situ* data (continuous colored lines) vs. theoretical prediction for  $\xi(z)$  (dashed red lines). Each panel shows the normalized depth  $(z-z_2)/H_2$  on the x-axis and the normalized isopycnal deviation  $(\xi-D)/\xi_0$  on the y-axis. The RMS values and eddy names are indicated in the panel captions. The colors correspond to those in Figure 8.

Figure 10. Comparison between the data (continuous colored lines) and the theoretical prediction for  $\eta_z(z)$  (dashed red lines). The horizontal axis represents the normalized depth,  $(z-z_1)/H_1$ , while the vertical axis depicts the normalized isopycnal deviation,  $(\eta_z - B)/\eta_{z0}$ . The root-mean-square (RMS) values and eddy names are provided in the caption. Colors correspond to those in Figure 8.

the deviations near the surface is the high variability in water properties near the surface due to atmospheric forcing, which influences both local stratification and the heat and salt content in the eddy core.

## 4.3.3 Vertical Extent of $\delta \sigma$

The vertical structure of  $\delta\sigma$  is examined by comparing the theoretical and observed values of  $\eta_z$ , as shown in Figure 10. The model closely fits the data, with RMS values consistently below 6% of the normalized amplitude. Figure 11 presents the normalized streamfunction, ,  $\Psi_v$ , obtained by integrating Equation 30. This function is used to determine the parameters  $\Psi_{v0}$ ,  $\mu$  and  $\zeta_0$  for the models of Flierl (1987) and Zhang et al. (2013). The corresponding parameter values are provided in Appendix A, Tables A3, A4, A5, A6. Once both  $\eta_z$  and  $\Psi_v$  are determined, we reconstruct  $\delta_z\sigma$  at the eddy center. The results are presented in Figure 12.

As illustrated in Figures 3 and 12, the model of Flierl (1987) assuming constant stratification produces RMS values frequently exceeding 15%, indicating limited accuracy. The same model, when adapted for variable stratification, performs better, with RMS values occasionally exceeding 10%, though it often reproduces the initial peak of the anomaly well. However, this peak corresponds to the shallower portion of the eddy, which is highly influenced by ocean surface fluctuations. The model of Zhang et al. (2013), despite yielding acceptable RMS values, fails to capture the shallow peak of the ACs sampled during the M124 cruise. In contrast, our proposed model provides the best overall agreement with observations, with RMS values consistently below 7%. Across all models, the density anomaly near the median plane is well captured, while the greatest discrepancies arise in the shallow layers of the eddies.

Figure 11. Normalized  $\Psi_v$  for each anticyclonic eddy as a function of the stretched coordinate  $\zeta$ . The coordinate  $\zeta$  is computed using the local stratification profiles presented in Appendix A, figure B1 and thus scales differently depending on the region.  $\Psi_v$  is obtained by integrating Equation 30.

#### 4.3.4 Two-Dimensional Vertical Reconstruction

In this final section, we present a comprehensive reconstruction of the potential density field using the Equation 48 model (Bara2024 model hereafter) and compare it with the observed *in situ* field. Since  $\delta^2 \sigma$  is negligible, it is omitted from the reconstructed field. To quantify the model's accuracy, the local relative error,  $\Delta$ , is computed as:

$$\Delta(x,z) = 100 \frac{|\sigma_{(insitu)}(x,z) - \sigma_{(model)}(x,z)|}{\sigma_{(insitu)}(x,z)}.$$
 (57)

Overall, the reconstruction yields an error of less than 5% in the core of the eddies, demonstrating a satisfactory level of precision. The error increases toward the eddy boundaries, particularly in regions with strong isopycnal slopes. For the surface AC sampled during EUREC4A-OA, the highest errors occur where the isopycnal slope is steepest. As shown by Barabinot et al. (2024), this region is influenced by small-scale turbulence, which can introduce minor modifications to the density field. The subsurface AC sampled during EUREC4A-OA is generally well represented by the model, although errors reach up to 15% in the shallower part of the eddy. This discrepancy coincides with the presence of small-scale structures, as discussed in Barabinot et al. (2024) and visible in panel (c) of Figure 8. For the AC sampled during Physindien 2011, the model does not fully capture the upper part of the vortex. As seen in Figure 12, the upper peak of the density anomaly is not well represented, likely due to the choice of stratification parameters (see Appendix B). For the three eddies sampled during M124, the model

Figure 12. Comparison between the observed *in situ heaving mode* density anomaly (dark lines) and the central eddy values of  $\delta_z \sigma$  predicted by the four proposed models: Flierl1987 equation 45 (continuous orange line) Flierl1987\_cst equation 46 (dashed orange line), Zhang2013 equation 47 (green line), Bara2024 equation 48 (blue line). The RMS error for each model is expressed as a percentage of the maximum *in situ* density anomaly.

performs well in the eddy cores, particularly near the median plane, confirming its robustness in representing the dominant density structures.

**Figure 13.** Relative error between the *in situ* data and Bara2024 model; expressed as a percentage. Panel titles indicate the analyzed eddy. Continuous dark lines represent *in situ* isopycnals for reference.

## 5 Discussion

## 5.1 Eddy Shape and Model Comparison

In contrast with previous studies (Bennani et al., 2022; Ayouche et al., 2021), the values of  $\alpha_1$  and  $\alpha_2$  in our analysis frequently exceed 2 (Gaussian eddies) and, in some cases, even 3 (cubic-exponential profiles). This parameter is critical as it governs the horizontal gradient of  $\delta\sigma$ , which in turn determines the velocity field. A higher  $\alpha_1$  results in a steeper horizontal gradient and increased velocity maximum. Among the six eddies analyzed, only two exhibit Gaussian profiles (Figures 8 (a) and 8 (d)). In the remaining cases, the density anomaly remains nearly constant in the core and decreases rapidly at the periphery. Such profiles may induce barotropic instabilities as previously observed by Carton and McWilliams (1989), though stratification stabilized the flow.

Gaussian shapes correspond to self-similar solutions associated with diffusive processes, as exemplified by the well-known Lamb-Oseen vortex in unsteady incompressible flow (Oseen, 1912). However, mesoscale eddies in the global ocean are rarely fully isolated. Their boundaries are subject to not only diffusion but also advection and shear from the background flow. This external forcing can erode the rotating flow, steepening the velocity profile and increasing  $\alpha_1$ . As a result, the vortex diffuses less momentum into the background flow (Legras and Dritschel, 1993; Mariotti et al., 1994). Vertically, Figures 3 and 12

show that the observed anomalies lack symmetry about the median plane, in contrast with vortices produced in laboratory experiments under constant stratification (Bonnier et al., 2000; Negretti and Billant, 2013; Mahdinia et al., 2017).

Figures 12 and 13 demonstrate that both models Flierl1987 and Bara2024 adequately capture the vertical structure of the potential density anomaly in the eddy cores. While model Bara2024 yields more accurate results, it introduces an additional parameter compared to model Flierl1987. The former is derived from self-similar diffusivity considerations, whereas the latter is based on quasi-geostrophic theory. Model Bara2024, supported by both experimental and observational studies, predicts that density anomalies vanish at depth, unlike model Flierl1987. However, model Flierl1987 is built on stronger theoretical foundations. Further investigation is needed to determine whether inertia or diffusion is the primary driver of coherent eddy shape.

Our study raises several important questions, many of which merit dedicated investigation in future work, including our own ongoing research. In particular, elucidating the physical interpretation of the parameters B and D, as well as the variability of B and B is essential for a better understanding of eddy dynamics.

## 5.2 Eddies Shape on Different Sections

This section presents additional vertical sections of eddies (a), (e), and (f) from Figure 3, analyzed using the same methodology.

The fitted models and reconstructed density fields using the Bara2024 model are shown in Figure 14. The low RMS errors indicate that the models successfully predict density variations in these additional sections.

While the horizontal profile maintains the same  $\alpha_1$  as in Figure 8, other parameter values (see Table A1) vary between sections. This suggests that the eddy shapes are not perfectly axisymmetric, despite appearing so in single-ship transects. This discrepancy in parameter values may arise from differences in the estimated eddy center across sections. In a single transect, the center is identified as the point where the orthogonal velocity is zero, but this estimate does not necessarily coincide with those obtained using Nencioli et al. (2008) or from another section. The influence of center misalignment is quantified in Appendix C for an idealized circular vortex.

Higher order azimuthal modes appear weak in coherent eddies, as indicated by the low RMS errors in the horizontal reconstruction of  $\delta\sigma$  in both sections of eddies (a), (e) and (f) in Figure 3. Although true circular vortices are rare, the radial profile remains largely unchanged when modeling slightly non-circular vortex. The key advantage of our model is its variable separation approach, commonly used for advection-diffusion equations, governing density anomalies. Introducing angular variations  $\theta$  does not fundamentally alter the reasoning. Figure 2 confirms that these eddies are not perfectly circular, yet our methodology and formulas accurately describe their structure (Figure 13). The variations of  $R_1$  across sections suggests that the horizontal shape of the density anomaly can be approximated by:

$$\phi(r,\theta) = \exp\left(-\left(\frac{r}{R_1(\theta)}\right)^{\alpha_1}\right),$$
 (58)

where the eddy radius  $R_1$  varies with the orientation  $\theta$  of the section.

Vertically, parameter values differ between sections (see Table A4). While the magnitudes remain consistent, notable discrepancies persist. These variations may result from differences in the estimated eddy center or the temporal gap between

**Figure 14.** Two-dimensional vertical reconstruction of eddies. Each column corresponds to one eddy and to one letter: (a) second subsurface AC cross section from EUREC4-OA experiment; (b) and (c) second AC cross sections from M124 experiment. The small map shows ship tracks (gray for sections in Figure 3, blue for the sections in this figure) with topography (Smith and Sandwell, 1997) and eddy centers (yellow dots) computed using (Nencioli et al., 2008) on gray sections. The second row presents the horizontal extent of  $\delta\sigma$  and the corresponding analytical model (as in Figure 8). The third row shows the relative error between the data and the two-dimensional reconstruction using the Bara2024 model (as in Figure 13).

section acquisitions. Since eddies can move or evolve during this period, vertical structure sampling may vary, as observed in previous studies (Flierl, 1981, 1987).

## 5.3 Impact of heaving and spiciness mode anomalies on geostrophy

As illustrated in Figures 7 and 5 and previously noted by (Lv et al., 2023), heaving mode anomalies are the dominant contributors to the total potential density anomaly in eddy cores. Consequently, the radial gradient of  $\delta\sigma$  is significantly larger than that of  $\delta^2\sigma$ , leading to stronger azimuthal geostrophic flow via thermal wind balance. Since mesoscale eddies are generally in geostrophic equilibrium (Carton, 2001; Carton et al., 2010), thermal wind is often the primary driver of horizontal rotational

flow. The flow generated by *heaving mode anomalies* therefore determines the intensity of the geostrophic velocity, the shape of the trapped water and, consequently, the horizontal structure of *spiciness mode anomalies*. This interaction explains why the characteristic parameters of both density anomaly types are nearly identical. While thermohaline anomalies of trapped water remain largely conserved during advection, their influence on the eddy's dynamics is minimal.

## 5.4 Discussion on cyclonic eddies

Previous studies have identified similar horizontal or vertical structures for mesoscale eddies of both polarities (Flierl, 1987; Chelton et al., 2011; Zhang et al., 2013). In the cruise analyzed, the lack of data does not permit us to study in detail the structure of cyclonic eddies. In the quasi-geostrophic framework, there should be no difference in the horizontal and vertical structures between cyclonic and anticyclonic eddies, except for the sign of their anomalies. The methodology and formulas proposed in this article can thus be applied to cyclonic eddies.

## 6 Conclusions

This study has examined the three-dimensional structure of anticyclonic eddies using *in situ* observations collected during oceanographic cruises. Unlike previous studies, which have primarily focused on the velocity field, our approach focused on the potential density field, which plays a fundamental role in sustaining the eddy's flow and coherence. The primary objective was to analyze the impact of *heaving* and *spiciness mode anomalies* on the potential density field within eddy cores and therefore on the induced geostrophic velocity that shapes them. Through a novel theoretical decomposition, of the potential density field, we confirm that *heaving mode anomalies* constitute the dominant contribution to the overall potential density anomaly, as previously suggested in the literature. However, our study expands on earlier findings by analyzing multiple eddy structures to assess this property more comprehensively.

A second key objective was to characterize the shape of both mode anomalies. A major contribution of this study was the proposal of two formulations to model the *heaving mode anomaly* in the eddy core: one based on a quasi-geostrophic framework, and the other derived from arguments inspired by experimental studies. By bridging oceanic observations and laboratory experiments, these formulations enhance our understanding of the physical mechanisms governing the eddy structure. They have practical implications for initializing eddy structures in numerical simulations, distinguishing between eddy shapes in model outputs, and estimating eddy volumes. Our findings also demonstrate that the vertical extent of the *heaving mode anomaly* is controlled by local stratification and confirm that eddies are not vertically symmetric about their median plane. The amplitude of the *heaving mode anomaly* is governed by the local background stratification and is thus greater at near surface than at depth. Because the background stratification is stronger near the surface, the isopycnals are more strongly deflected in the lower part of the anticyclonic eddy than in its upper part.

In summary, this study provides new insights into the three-dimensional structure of anticyclonic mesoscale eddies. A similar investigation of cyclonic eddies would be a valuable next step. Further quantitative validation is required to confirm the broader

applicability of these results. Accurately determining eddy volumes remains crucial for assessing their role in heat, carbon, and tracer transport.

Data availability. This study, we benefited from multiple freely available datasets, listed below:

The concatenated hydrographic and velocity data *Atalante* and *Maria S. Merian* (L'Hégaret Pierre, 2020) are available on SEANOE: https://doi.org/10.17882/92071 (accessed on 15 March 2021).

Hydrographic and velocity measurements from the M124 cruise (Karstensen and Wölfl, 2016; Karstensen et al., 2016b; Karstensen and Krahmann, 2016) aboard the RV Meteor are available on PANGAEA:

https://doi.org/10.1594/PANGAEA.902947, https://doi.pangaea.de/10.1594/PANGAEA.863015, https://doi.pangaea.de/10.1594/PANGAEA.590 869740.

Hydrographic and velocity measurements from the Physindien 2011 campaign (L'Hégaret and Carton, 2011; L'Hégaret et al., 2016) are available on SEANOE:

https://doi.org/10.17882/77351.

## **Appendix A: Parameters Values**

The following tables present the parameters values used in this study. Abreviations "EUR", "Phy", "surf", "sub", refer to "EUREC4A-OA", "Physindien 2011", "surface", "subsurface", respectively.

**Table A1.** Parameters values for  $\phi(x)$  and  $\chi(x)$ 

| Eddy               | $x_1[\mathrm{km}]$ | $R_1[\mathrm{km}]$ | $x_2[\mathrm{km}]$ | $R_2[\mathrm{km}]$ |
|--------------------|--------------------|--------------------|--------------------|--------------------|
| AC surf EUR        | 184                | 94                 | 187                | 97                 |
| AC sub EUR         | 99                 | 88                 | 98                 | 85                 |
| AC sub EUR (sec 2) | 280                | 81.9               | -                  | -                  |
| AC surf Phy        | 462                | 56                 | 461                | 49                 |
| AC1 M124           | 72                 | 50                 | 72                 | 43                 |
| AC2 M124           | 48                 | 70                 | 50                 | 81                 |
| AC2 M124 (sec 2)   | 797                | 96.9               | -                  | -                  |
| AC3 M124           | 131                | 102                | 130                | 100                |
| AC3 M124 (sec 2)   | 437                | 95.5               | -                  | -                  |

## **Appendix B: Stratification**

Figure B1 presents local stratification profiles for studied eddies.

**Table A2.** Parameters values for  $\delta_z^2(z)$ 

| Eddy        | $z_2[\mathrm{m}]$ | $H_2[\mathrm{m}]$ | $\xi_0  [{\rm kg.m}^{-3}]$ | D[m]     |
|-------------|-------------------|-------------------|----------------------------|----------|
| AC surf EUR | -175              | 62                | -3.02e-2                   | -1.46e-2 |
| AC sub EUR  | -254              | 102               | -3.77e-3                   | -9.98e-3 |
| AC surf Phy | -118              | 101               | -8.50e-3                   | -1.49e-1 |
| AC1 M124    | -121              | 113               | 6.40e-2                    | -1.48e-1 |
| AC2 M124    | -127              | 83                | 6.70e-2                    | -1.48e-1 |
| AC3 M124    | -109              | 111               | 8.60e-2                    | -1.49e-1 |

**Table A3.** Parameters values for  $\eta_z(z)$ 

| Eddy               | $z_1[\mathrm{m}]$ | $H_1[m]$ | $\eta_{z0}[\mathrm{m}]$ | B[m]  |
|--------------------|-------------------|----------|-------------------------|-------|
| AC surf EUR        | -55               | 139      | 146                     | 2     |
| AC sub EUR         | -343              | 385      | 276                     | -72   |
| AC sub EUR (sec 2) | -318              | 328      | 227                     | -52   |
| AC surf Phy        | -63               | 236      | 187                     | -3    |
| AC1 M124           | -243              | 157      | 103                     | -6    |
| AC2 M124           | -226              | 126      | 140                     | -8    |
| AC2 M124 (sec 2)   | -210              | 143      | 199                     | -63.8 |
| AC3 M124           | -200              | 99       | 78                      | 4     |
| AC3 M124 (sec 2)   | -186              | 154      | 152                     | -45   |

**Table A4.** Parameters values for  $\Psi_v(z)$  (Flierl, 1987)

| Eddy        | $\Psi_{v0}  [\mathrm{m}^2.\mathrm{s}^{-1}]$ | $\mu  [\mathrm{m}^{-1}]$ | $\zeta_0[\mathrm{m}]$ |
|-------------|---------------------------------------------|--------------------------|-----------------------|
| AC surf EUR | -2.45e3                                     | 9.78e-6                  | 1.16                  |
| AC sub EUR  | 9.61e2                                      | 4.43e-5                  | 4.28                  |
| AC surf Phy | 6.36e2                                      | 6.69e-5                  | -2.15                 |
| AC1 M124    | 1.90e2                                      | 1.92e-4                  | -2.30                 |
| AC2 M124    | -1.95e2                                     | 2.30e-4                  | -6.24                 |
| AC3 M124    | 1.55e2                                      | -1.52e-4                 | -4.88                 |

Appendix C: Link between Vertical Sections and Three-Dimensional Structure

Determining the three-dimensional structure of eddies, particularly their horizontal extent, based solely on two-dimensional vertical sections remains challenging. Horizontal reconstructions from vertical sections require extrapolation and rely on specific assumptions about the eddy's geometry.

**Table A5.** Parameters values for  $\Psi_v(z)$  (Flierl, 1987) for a constant stratification

| Eddy        | $\Psi_{v0}  [\mathrm{m}^2.\mathrm{s}^{-1}]$ | $\mu[\mathrm{m}^{-1}]$ | $\zeta_0  [\mathrm{m}]$ |
|-------------|---------------------------------------------|------------------------|-------------------------|
| AC surf EUR | -2.58e4                                     | 2.21e-5                | -6.05e-1                |
| AC sub EUR  | -8.15e3                                     | 1.33e-5                | 3.50e-1                 |
| AC surf Phy | 7.90e3                                      | -6.35e-5               | 2.70                    |
| AC1 M124    | -1.09e3                                     | -7.36e-5               | 9.30e-2                 |
| AC2 M124    | 1.29e3                                      | 8.57e-5                | -4.21                   |
| AC3 M124    | -3.54e3                                     | 9.67e-5                | -8.72e-1                |

**Table A6.** Parameters values for  $\Psi_v(z)$  (Zhang et al., 2013)

| Eddy        | $\Psi_{v0} \ [\mathrm{m^2.s^{-1}}]$ | $\mu[\mathrm{m}^{-1}]$ | $\zeta_0$ [m] |
|-------------|-------------------------------------|------------------------|---------------|
| AC surf EUR | 2.56e4                              | -3.24e-5               | 3.03          |
| AC sub EUR  | 1.19e4                              | -3.84e-5               | 2.03          |
| AC surf Phy | 9.55e3                              | -6.75e-5               | 2.21          |
| AC1 M124    | -1.21e3                             | -2.12e-4               | 5.79          |
| AC2 M124    | -1.39e3                             | -2.52e-4               | 6.63          |
| AC3 M124    | -3.55e2                             | -3.66e-4               | 7.77e-1       |

**Figure B1.** (Left) Climatological profiles of the potential density following the methodology of Section 3.3. (Right) Associated Brunt-Väisälä frequency

This section examines the relationship between the representation of an eddy in a two-dimensional vertical section and its actual three-dimensional cylindrical structure.

For simplicity, we assume the eddy is axisymmetric and that the ship track perfectly intersects its center. In this idealized case, the horizontal coordinate r corresponds directly to x, and the horizontal functions  $\phi$  and  $\chi$  are perfectly aligned. However, real eddies are often non-axisymmetric (Chen et al., 2019), complicating the inference of azimuthal variations from a single ship track. Incorporating non-axisymmetry would introduce coupling between the radial (r) and azimuthal  $(\theta)$  coordinates, complicating the calculations. Here, we focus on estimating the order of magnitude of the error.

Figure C1. Schematic representation of a ship track (red squares) crossing an axisymmetric eddy of radius  $R_1$  with a small lateral displacement e. The eddy radius appears as  $R_x < R_1$  in the vertical section.

In practice, the optimal ship track rarely passes precisely through the center of an eddy. A small horizontal displacement of the track, denoted as e (see Figure C1), results in a misalignment between the horizontal coordinate in the ship's reference frame (x) and the true radial coordinate (r) of the eddy. Consequently, understanding the impact of this displacement on two-dimensional vertical sections is essential.

Since potential density anomalies are strongest at the eddy center and decrease outward, any displacement of the ship track affects the observed structure of these anomalies. This shift alters the horizontal functions  $\phi$  and  $\chi$ , which characterize the horizontal density anomaly profiles. Figure C1 illustrates the relationship between the true three-dimensional horizontal functions,  $\phi_r$  (or  $\chi_r$ ), and their counterparts as observed in the two-dimensional vertical section,  $\phi_x$  (or  $\chi_x$ ). The objective is to quantify the error introduced when an eddy is sampled along a misaligned section.

#### C1 Quantifying the Error in the Observed Profile

We begin with the horizontal function  $\phi_r$ , which in three dimensions is expressed as:

$$\phi_r(r) = \exp\left(-\left(\frac{r}{R_1}\right)^{\alpha_1}\right). \tag{C1}$$

Using the coordinate transformation  $r^2 = x^2 + e^2$  and defining the effective radius in the vertical section as  $R_x$  such that  $R_x^2 + e^2 = R_1^2$ , we can rewrite  $\phi_r$  in terms of x:

$$\phi_r(x) = \exp\left(-\left(\frac{x^2 + e^2}{R_x^2 + e^2}\right)^{\alpha_1/2}\right).$$
 (C2)

For comparison, the function  $\phi_x$  directly obtained from the vertical section is:

$$\phi_x(x) = \exp\left(-\left(\frac{x}{R_x}\right)^{\alpha_1}\right). \tag{C3}$$

Since the parameter  $\alpha_1$  remains unchanged under vertical sectioning, we analyze the impact of the displacement e by considering three limiting cases:

## Case 1: Near the Eddy Center $(x \approx 0, r \approx e)$

At x = 0, the function  $\phi_r$  simplifies to:

$$\phi_r(e) = \exp\left(-\left(\frac{e^2}{R_x^2 + e^2}\right)^{\alpha_1/2}\right). \tag{C4}$$

Rewriting it in terms of  $e/R_x$ :

$$\phi_r(e) = \exp\left(-\left(\frac{e}{R_x}\right)^{\alpha_1} \left(\frac{1}{1 + (e/R_x)^2}\right)^{\alpha_1/2}\right). \tag{C5}$$

For small  $e/R_x$ , applying a Taylor expansion yields:

$$\phi_r(e) = \phi_x(e) \left( 1 + \frac{\alpha_1}{2} \left( \frac{e}{R_x} \right)^{\alpha_1 + 2} + O\left( \left( \frac{e}{R_x} \right)^{\alpha_1 + 2} \right) \right).$$
 (C6)

Thus, at the eddy center, the observed function  $\phi_x$  is only slightly altered by the displacement e.

# Case 2: Intermediate Displacement $(x \approx e \Leftrightarrow r \approx \sqrt{2}e)$

For  $x \approx e$ , which corresponds to a radial coordinate  $r \approx \sqrt{2}e$ , we write:

$$\phi_r(\sqrt{2}e) \approx \exp\left(-\left(\frac{2e^2}{R_x^2 + e^2}\right)^{\alpha_1/2}\right).$$
 (C7)

Using a Taylor expansion in terms of  $e/R_x$ , we obtain:

$$\phi_r(\sqrt{2}e) = \phi_x(\sqrt{2}e) \left( 1 + (\sqrt{2})^{\alpha_1} \frac{\alpha_1}{2} \left( \frac{e}{R_x} \right)^{\alpha_1 + 2} + O\left( \left( \frac{e}{R_x} \right)^{\alpha_1 + 2} \right) \right). \tag{C8}$$

# Case 3: Large Displacement $(x \gg e \Leftrightarrow r \approx x)$

For sufficiently large x (i.e., further from the eddy center), we approximate:

$$\phi_r(x) \approx \exp\left(-\left(\frac{x^2}{R_x^2 + e^2}\right)^{\alpha_1/2}\right).$$
 (C9)

Expanding in terms of  $e/R_x$  gives:

$$\phi_r(x) = \phi_x(x) \left( 1 + \frac{\alpha_1}{2} \left( \frac{e}{R_x} \right)^2 \left( \frac{x}{R_x} \right)^{\alpha_1} + O\left( \left( \frac{e}{R_x} \right)^2 \right) \right). \tag{C10}$$

## **C2** Error Estimation and Implications

Since x is generally of the order of  $R_x$ , the overall error due to displacement e can be expressed as:

Error 
$$\sim \alpha_1 \left(\frac{e}{R_x}\right)^2$$
. (C11)

Key observations:

- Quadratic Dependence on Displacement: The error grows as  $e^2$ , meaning that a larger misalignment leads to a more significant deviation from the actual eddy structure.
- Effect of Steepness Parameter  $\alpha_1$ : The steeper the density anomaly profile (higher  $\alpha_1$ ), the more sensitive it is to misalignment.
- **Dependence on Eddy Size:** Larger eddies (higher  $R_x$ ) are less affected by displacement errors, as their structure is more resilient to minor shifts.

For illustration, consider a Gaussian eddy ( $\alpha_1 = 2$ ) with a radius of 100 km. If a ship track is displaced by 10 km, the resulting error in  $\phi_x$  is approximately 1%, which is relatively minor.

These findings emphasize the importance of precise eddy center localization in *in situ* sampling. Even small shifts in the ship track can influence the observed density structure, particularly for smaller or sharply defined eddies

Author contributions. All the authors contributed to conception and design of the study. YB wrote the first draft of the manuscript. All authors contributed to the article and approved the submitted version.

Competing interests. The authors report no conflict of interest.

Acknowledgements. The authors acknowledge funding from the European Research Council (ERC) under the European Union's Horizon 2020 research and innovation programme (grant agreement No 101118693 — WHIRLS) and from the European Union's Horizon Europe research and innovation programme (grant agreement No 101136203 — ObsSea4Clim), the Centre National d'Études Spatiales (CNES) through the TOEddies and EUREC4A-OA projects, the French National Program LEFE INSU, IFREMER, the French Vessel Research Fleet, the DATA TERRA French Research Infrastructures AERIS and ODATIS, IPSL, the Chaire Chanel Program of the ENS Geosciences Department, and the EUREC4A-OA JPI Ocean and Climate Program. We extend our gratitude to all individuals who contributed to the collection, processing, and public dissemination of the data, as well as to the institutions they represented, particularly the University of Bergen and GEOMAR Helmholtz Centre for Ocean Research Kiel. We also express our sincere appreciation to the captains and crews of the RVs Atalante, Maria S. Merian, FS Meteor, whose efforts were instrumental in the successful completion of this study. YB acknowledges

financial support from a Ph.D. grant provided by the École Normale Supérieure Paris Saclay. XC acknowledges support from Université de Bretagne Occidentale (UBO) and a CNES contract under the EUREC4A-OA program.

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
