# Peer review of "Insights Into Mesoscale Eddy Dynamics: A Three-Dimensional Perspective on Potential Density Anomalies"

_EGUsphere, 2025_

## Referee Comment (RC1)

**Review of Insights Into Mesoscale Eddy Dynamics: A Three-Dimensional Perspective on Potential Density Anomalies**

**1  Summary**

This manuscript investigates the 3D structure of mesoscale eddies based on in situ observations and theories. The authors decompose the eddy density anomaly into the spiciness mode and heaving mode, finding the heaving mode to be dominant. They further evaluate the vertical and horizontal structures of eddy density field in light of the quasigeostrophic and diffusion theories, concluding that the diffusion theory effectively captures the observed 3D eddy structure. Overall, I find the analysis and discussion to be thorough and insightful. The approach of decomposition into spiciness and heaving modes is novel, and the proposed diffusion theory offers valuable perspectives on the vertical structure of eddy density anomalies. However, several aspects of the analysis and interpretation require further clarification to strengthen the overall argument. I recommend a major revision addressing the following comments.

**2  Major Comments**

1. This manuscript focuses primarily on anticyclonic eddies and leaves the 3D structure of cyclonic eddies mysterious. I do not think that the dynamics underlying cyclonic and anticyclonic eddies are fundamentally different. In fact, previous studies (including those cited in the manuscript) have identified similar horizontal or vertical structures for mesoscale eddies of both polarities (Flierl, 1987; Chelton et al., 2011; Zhang et al., 2013). In the QG framework, there should be no difference in the horizontal and vertical structures between cyclonic and anticyclonic eddies, except for the sign of their anomalies. While the dataset used here only includes anticyclonic eddies, I suggest adding a discussion of whether and how the theories can be applied to cyclonic eddies. Would any adjustment be needed to extend the theoretical framework to account for cyclonic eddies?

2. The theoretical formulations for the eddy 3D structure (eq. (53)-(56)) contain 14 unknown parameters, which were estimated through fitting. Some parameters, such as B and D in equations (55) and (56), do not have a clear physical meaning. When the observation is sparse, such as Argo observations, it is unlikely to conduct the fit to estimate those parameters. Can any of the parameters be estimated directly from limited observations? For example, if $R_1$ and $R_2$ in equations (53) and (54) represent the radii of eddies, could they be estimated from satellite altimetry using an approach similar to that in Chelton et al. (2011)? Could $x_1$ and $x_2$ be estimated directly as the location of the maximum surface density anomaly rather than by fitting? The fitted values of $R$ and $H$ are different at different regions. Does this difference reflect physical characteristics of the local environment, such as the Rossby deformation radius and e-folding depth of stratification?

3. It is unclear how the parameters in equations (53)-(56) are estimated through optimization. The manuscript briefly mentions that the optimization is conducted for the vertical structure first and then for the horizontal structure, but it remains unclear how the horizontal and vertical structures are obtained from the observational data. For instance, the vertical structure function is said to be optimized at the eddy center. Is the eddy center at $x_1$ or $x_2$? What if the location of eddy center changes with depth when an eddy tilts vertically? Furthermore, is the vertical structure, when normalized by its maximum, consistent at different radial distances from the eddy center? In addition, Zhang et al. (2013) proposed a universal horizontal structure function for mesoscale eddies (their equation (2)), which includes a sign reversal of eddy pressure anomalies, beyond approximately 1.4 times the eddy radius (their figure 2). However, this feature does not appear to be present in figure 8 of the manuscript. I suggest comparing the observed horizontal structure to the function proposed by Zhang et al. (2013) to provide a better justification for the proposed framework.

**3  Minor Comments**

1. Equation (53) and (54): Is the $\alpha$ enforced to be an integer? It seems that if $\alpha$ is odd and $x - x_1$ is negative, the eddy anomaly will increase exponentially from its center, which is unphysical. Perhaps, the absolute value should be taken for $x - x_1$?

2. I am not sure whether the tilde above all variables is necessary. It seems that it does not have a special meaning and is in fact dropped in later sections. To make the derivations clearer, I suggest dropping tilde completely.

3. Line 119: I think $\eta$ should be the vertical displacement of isopycnal with respect to the mean state, not the state of rest.

4. Line 134: I suggest dropping "we assume". You can estimate the typical density variation corresponding to the isopycnal height change, not subjectively assuming it.

5. Lines 141 and 170: Missing spaces between $\tilde{\cdot}$ and texts.

6. Equation (21): What's the difference between $T(r, \theta, \sigma_0)$ and $\tilde{\bar{T}}(\sigma_0)$? It feels that equation (21) is just zero equals zero.

7. Line 185: I do not think it is accurate to say "no exact analytical expression for $\phi(r)$ exists". Zhang et al. (2013) has proposed an analytical expression for the radial structure of eddies.

8. Line 208: "radius" → "radial distance".

9. Line 223: I suggest drop "assuming" for the same reason mentioned before.

10. Line 248: $H_1$ is a characteristic **depth** scale.

11. Table 1: A "vertical" is not capitalized.

12. Line 268: How exactly is the eddy center be determined from velocity analysis?

13. Table 2: The filter scale for x is different for different observations. What's the justification for the filter scale? Is it based on the local deformation radius?

14. Line 319 and others: "cutoff period" → "cutoff scale" or "filter scale".

15. Line 367: $\psi_0$ should be $\eta_{z0}$.

16. Line 373: "$\psi$ and $\xi$ " → "$\eta_z$ and $\delta_z^2 \sigma$"? I see no $\psi$ and $\xi$ in equations (53)-(56).

17. Line 374: There is an extra period.

18. Line 374: "locations where the amplitude of $\psi$ and $\xi$ are maximal" → "depth where the amplitude of $\eta_z$ and $\delta_z^2 \sigma$ are maximal"?

19. Line 389: Why is the water in an anticylconic eddy be colder than its surroundings?

20. Figure 8: Which depth are the $\phi$ and $\chi$ from? The depth where $\eta_z$ is the maximum? Is $\phi(x)$ or $\chi(x)$ different at other depth?

21. Line 441: There is an extra "function".

22. Figures 9 and 10: Which location are the vertical profiles from? The eddy center? Is $\eta_z(z)$ or $\delta_z^2 \sigma(z)$ different at different x?

23. Line 465: The theory here has more tuning parameters than the models of Flierl (1987) and Zhang et al. (2013). The comparison between them seems to be unfair.

24. Line 508: "reconstruct" → "reconstructed".

25. Figure 14: Labels a, b, and c are repeated in the second row, and there is no alphabetic label in the third row. Please correct them.

26. Line 549: "diffusivity" → "diffusion".

27. Line 553: There should be more details about how the vertical extent is controlled by the stratification and what the vertical symmetry means.

**References**

Chelton, D. B., M. G. Schlax, and R. M. Samelson, 2011: Global observations of nonlinear mesoscale eddies. *Prog. Oceanogr.*, **91 (2)**, 167–216.

Flierl, G., 1987: Isolated eddy models in geophysics. *Annu. Rev. Fluid Mech.*, **19 (1)**, 493–530.

Zhang, Z., Y. Zhang, W. Wang, and R. Huang, 2013: Universal structure of mesoscale eddies in the ocean. *Geophys. Res. Lett.*, **40**, 3677–3681.

---

## Author Comment (AC1)

**Reply referee 2**

We would like to thank the reviewer for his/her review and constructive feedback. We appreciate the effort and time the reviewer has invested in evaluating our work. Please find our point-to-point response below in blue.

*I found this to be a useful contribution to the literature on the structure of coherent anticyclonic eddies in the ocean. The authors essentially fit idealized horizontal and vertical structure functions to five coherent anticyclonic eddies. A key novelty, compared with earlier studies are the use of potential density, rather than velocity, fields in fitting the models to the observations. The authors find that heaving anomalies dominate over spiciness anomalies - which is perhaps not surprising, but nevertheless valuable to confirm. There is also an interesting finding regarding the horizontal structure of the eddies, where the exponential power is higher than Gaussian or even cubic exponential - indeed I thought that this point could have been highlighted in the conclusions. Overall, the manuscript is well written and should be published after minor revisions.*

*I note that the first reviewer has commented extensively on the parameter fitting/optimization process and, in particular, number of free parameters as well as the detailed methodology. I will not dwell on these issues here, on which I am not as expert, except to note that I agree with the points raised by the reviewer.*

*Specific points:*

*1. The first few sentences of the abstract need tightening. Firstly, I disagree with the statement "In situ observations and Lagrangian analyses have shown that most eddies are materially coherent", indeed the work of Abernathey and Haller (doi: 10.1175/JPO-D-17-0102.1) reaches a very different conclusion. The easiest solution, since this point is of tangental relevance to the present manuscript, is to revert to the text in the introduction about coherent vortices being long-lived and playing an important role in ocean circulation and transport. There is also the separate issue raised by Abernathey and Haller of whether "eddy" is a noun or adjective, which again I would encourage the authors to sidestep by inserting the adjective "coherent".*

We thank the reviewer for this insightful remark. The concept of "coherence" is indeed complex, and there is ongoing debate within the community, particularly between proponents of Lagrangian and Eulerian approaches. While we agree that the precise proportion of "coherent" eddies remains uncertain, we respectfully disagree with the assertion from Abernathey and Haller that coherent eddies constitute only a minority. Most of Lagrangian studies use satellite-derived geostrophic fields, which are inherently two-dimensional and smoothed, to evaluate eddy coherence.. However, as illustrated in high-resolution models (e.g., Gula et al., 2022, see figure below), the ocean exhibits much greater complexity and chaos than what is captured by traditional altimetry. . The outcome of applying Lagrangian criteria to such realistic model simulations remains an open question. Furthermore, as shown by Liu et al. (2019), the number of coherent eddies identified depends strongly on the integration timescale chosen for the Lagrangian analysis..

In our view, it is perhaps too strong to claim that Lagrangian criteria applied to altimetry provide a definitive assessment of eddy coherence. We are equally critical of the application of Eulerian criteria to altimetric data. Our intent in this comment is not to question the value of Lagrangian criteria themselves, but rather to highlight the limitations of the observational tools commonly used to assess coherence.

In contrast, observational studies often suggest a higher proportion of coherent eddies. Materially coherent eddies are capable of transporting water masses over significant distances. Comparing water mass properties between eddy cores and their environment is also a valid method for assessing material coherence (Barabinot et al., 2025). Using this approach, previous studies have shown that eddies with thermohaline anomalies are not uncommon—for example, meddies (Armi et al., 1989), North Brazil Current rings (Barabinot et al., 2024), Agulhas rings (Laxenaire et al., 2019, 2020), Arabian Sea eddies (de Marez et al., 2020), and eddies in the tropical Atlantic (Aguedjou et al., 2021), among others.

In conclusion, we recognize that the wording "most of" may have been too strong, and we replaced it with "some" in the revised version. However, we maintain that coherent eddies are not necessarily a minority, as suggested by some Lagrangian studies.

[Figure]

Figure 1. Surface relative vorticity normalized by the Coriolis parameter in the Gulf stream in a very high resolution numerical model simulation  Gula et al. (2022) (dx=500 m)

*Secondly, I don't understand the statement "laboratory experiments indicate that eddies locally modify stratification in accordance with thermal wind balance, regardless of whether they trap a water mass". It is not obvious that stratification needs to altered at all in a heaving mode, but merely that the isopycnals and raised or lowered. And at low Rossby number, away from boundaries, any flow should be close to thermal wind balance, but I don't believe it follows that this causes eddies to modify stratification, only that lateral density gradients and vertical shear will co-vary in accordance with thermal wind balance.*

Our intention was to convey that an eddy locally deflects isopycnal surfaces, thereby altering the vertical gradient of buoyancy (and also the horizontal gradient of buoyancy). We have clarified this sentence in the revised manuscript.

*2. I understand that it is convenient to use potential density referenced to the surface, but is there a good reason not to have used neutral density for this study, given that the amount of data involved does not seem particularly prohibitive?*

We thank the reviewer for his/her relevant comment. As noted, we used the potential density for convenience. Since the eddies considered in this study are confined to the upper ocean layer, the difference between neutral density and potential density is negligible.

*3. Line 84: I think you need to add that the stratification is assumed to be stable for invertibility.*

We thank the reviewer for his/her relevant suggestion. A clarifying sentence has been added in the revised manuscript (see line 83).

*4. Figure 1: I suggest reducing the magnitude of the sea surface displacement in the upper panel.*

We thank the reviewer for this suggestion. The magnitude of the sea surface displacement in the upper panel has been reduced in the revised manuscript.

*5. I understand the rationale for invoking the quasigeostrophic approximation for analytical tractability, but you should comment on the extent to which the quasigeostrophic assumptions are (not) satisfied in the eddies under consideration, and whether this is a significant limitation, in your view, or otherwise (and why).*

We thank the reviewer for his/her suggestion. Additional comments have been included in the revised manuscript (see lines 211-216).

*6. I must confess I spent some time to derive equation (38) and wonder if you can provide some pointers to help the reader? Also, is there a good reason that the zero is moved to the left hand side of (37) compared with the original equation (34)?*

We thank the reviewer for his/her remark. To assist the reader, we have added an additional step in the derivation of equation (38) in the revised manuscript (see line 241). In addition, we have corrected equation (37).

*7. Line 215: I understand what you mean by the "nonlinear" term, but technically this remains a linear equation. It is nonlinear only in the vertical coordinate.*

We thank the reviewer for bringing this to our attention. We have proposed an alternative formulation in the revised manuscript (see line 233).

*8. Diffusion arguments" - what specifically do you mean by diffusion? Is this physical, based on an oceanographic process? I have no idea from what is written, and this must be explained and justified.*

We thank the reviewer for his/her relevant comment. In experimental studies, the diffusion of momentum governs the self-similar profile of the density anomaly in eddies, which explains why eddies often exhibit a Gaussian shape. Therefore, we sought a formula consistent with this observation. To clarify this point, we have modified the subsection title to "Arguments based on experimental studies" in the revised manuscript.

References

Aguedjou, H. M. A., Chaigneau, A., Dadou, I., Morel, Y., Pegliasco, C., Da-Allada, C. Y., & Baloïtcha, E. (2021). What can we learn from observed temperature and salinity isopycnal anomalies at eddy generation sites? Application in the tropical Atlantic Ocean. Journal of Geophysical Research: Oceans, 126(11), e2021JC017630.

Armi, L., Hebert, D., Oakey, N., Price, J. F., Richardson, P. L., Rossby, H. T., & Ruddick, B. (1989). Two years in the life of a Mediterranean salt lens. Journal of Physical Oceanography, 19(3), 354-370.

Barabinot, Y., Speich, S., & Carton, X. (2024). Defining mesoscale eddies boundaries from in-situ data and a theoretical framework. Journal of Geophysical Research: Oceans, 129(2), e2023JC020422.

Barabinot, Y., Speich, S., & Carton, X. (2025). Assessing the thermohaline coherence of mesoscale eddies as described from in situ data. Ocean Science, 21(1), 151-179.

de Marez, C., Carton, X., Corréard, S., L'Hégaret, P., & Morvan, M. (2020). Observations of a deep submesoscale cyclonic vortex in the Arabian Sea. Geophysical Research Letters, 47(13), e2020GL087881.

Jonathan Gula, John Taylor, Andrey Shcherbina, Amala Mahadevan, Chapter 8 - Submesoscale processes and mixing, Ocean Mixing, Elsevier, 2022, Pages 181-214, ISBN 9780128215128, https://doi.org/10.1016/B978-0-12-821512-8.00015-3.

Laxenaire, R., Speich, S., & Stegner, A. (2019). Evolution of the thermohaline structure of one Agulhas ring reconstructed from satellite altimetry and Argo floats. Journal of Geophysical Research: Oceans, 124(12), 8969-9003.

Laxenaire, R., Speich, S., & Stegner, A. (2020). Agulhas ring heat content and transport in the South Atlantic estimated by combining satellite altimetry and Argo profiling floats data. Journal of Geophysical Research: Oceans, 125(9), e2019JC015511.

Liu, T., Abernathey, R., Sinha, A., & Chen, D. (2019). Quantifying Eulerian eddy leakiness in an idealized model. Journal of geophysical research: Oceans, 124(12), 8869-8886.

---

## Author Comment (AC2)

**Reply referee 1**

We would like to thank the reviewer for his/her review and constructive feedback. We appreciate the effort and time the reviewer has invested in evaluating our work. Please find our point-to-point response below in blue.

**1 Summary**

*This manuscript investigates the 3D structure of mesoscale eddies based on in situ observations and theories. The authors decompose the eddy density anomaly into the spiciness mode and heaving mode, finding the heaving mode to be dominant. They further evaluate the vertical and horizontal structures of eddy density field in light of the quasigeostrophic and diffusion theories, concluding that the diffusion theory effectively captures the observed 3D eddy structure. Overall, I find the analysis and discussion to be thorough and insightful. The approach of decomposition into spiciness and heaving modes is novel, and the proposed diffusion theory offers valuable perspectives on the vertical structure of eddy density anomalies. However, several aspects of the analysis and interpretation require further clarification to strengthen the overall argument. I recommend a major revision addressing the following comments.*

**2 Major Comments**

*1. This manuscript focuses primarily on anticyclonic eddies and leaves the 3D structure of cyclonic eddies mysterious. I do not think that the dynamics underlying cyclonic and anticyclonic eddies are fundamentally different. In fact, previous studies (including those cited in the manuscript) have identified similar horizontal or vertical structures for mesoscale eddies of both polarities (Flierl, 1987; Chelton et al., 2011; Zhang et al., 2013). In the QG framework, there should be no difference in the horizontal and vertical structures between cyclonic and anticyclonic eddies, except for the sign of their anomalies. While the dataset used here only includes anticyclonic eddies, I suggest adding a discussion of whether and how the theories can be applied to cyclonic eddies. Would any adjustment be needed to extend the theoretical framework to account for cyclonic eddies?*

We thank the reviewer for this relevant comment. We agree that, in principle, the dynamics of cyclonic and anticyclonic eddies should not be fundamentally different, except for the sign of their anomalies. Consequently, there should be no systematic difference in their horizontal or vertical structures. The reason cyclonic eddies are not analyzed in detail in our study is solely due to the lack of suitable in situ data. Specifically, only a few cyclonic eddies were identified in our cruise dataset, and those that were observed were either sampled with insufficient resolution or only partially sampled (e.g., cross-sections did not capture the full vertical structure). This limitation is therefore a matter of data quality rather than an underlying physical difference. We have clarified this point and included a dedicated paragraph in the revised version of the manuscript (see Section 5.4 in the Discussion).

*2. The theoretical formulations for the eddy 3D structure (eq. (53)-(56)) contain 14 unknown parameters, which were estimated through fitting. Some parameters, such as B and D in equations (55) and (56), do not have a clear physical meaning. When the observation is*

*sparse, such as Argo observations, it is unlikely to conduct the fit to estimate those parameters. Can any of the parameters be estimated directly from limited observations? For example, if R1 and R2 in equations (53) and (54) represent the radii of eddies, could they be estimated from satellite altimetry using an approach similar to that in Chelton et al. (2011)? Could x1 and x2 be estimated directly as the location of the maximum surface density anomaly rather than by fitting? The fitted values of Rand H are different at different regions. Does this difference reflect physical characteristics of the local environment, such as the Rossby deformation radius and e-folding depth of stratification?*

We thank the reviewer for this constructive remark. We agree our study raises several important questions, many of which merit dedicated investigation in future work, iincluding our own ongoing research. Please find below additional details and hypotheses in response to the reviewer's queries:

- *Application to Argo Data:* We have recently conducted a Master's project to test whether our formulas can be applied to characterize eddies using Argo float profiles. The results are promising (including for the estimation of B and D), although they are not yet published. his demonstrates that Argo data can indeed be used to extend our analysis of eddy vertical structure.
- Physical Interpretation of B and D: Our working hypothesis is that the parameters B and D are linked to baroclinic vertical modes, but this remains to be confirmed with further analysis.
- Comparison of $R_1$ and $R_2$ from Satellite and In Situ Data: The question regarding R1 and R2, the question raised can be addressed by comparing satellites observations with in situ measurements which is feasible and could form the basis of a future study. In our present work, assuming $R_1$ and $R_2$ to be constant was sufficient to model the horizontal structure of eddies at various depth levels. However, this assumption may not hold in all cases, suggesting that satellite data alone may not always be adequate.
- Interpretation of $x_1$ and $x_2$: As shown in Table A1, x1 is approximately equal to x2, representing the average eddy center location on cross-sections. At the surface, it thus makes sense to estimate x1 and x2 at the location of maximum surface density anomaly. If the eddy is drifting, x1 and x2 would be expected to vary with depth.
- Variability of R and H: Understanding the spatial variability of R and H, as well as their temporal evolution throughout the eddy lifecycle, is indeed crucial. The main objective of our article was to propose and validate formulas to characterize eddy structure at specific moments ("snapshots"). However, a comprehensive analysis of their space–time variability will require further dedicated research.

We have clarified this point and included a dedicated paragraph in the revised version of the manuscript between lines 516 and 520.

*3. It is unclear how the parameters in equations (53)-(56) are estimated through optimization. The manuscript briefly mentions that the optimization is conducted for the vertical structure first and then for the horizontal structure, but it remains unclear how the horizontal and vertical structures are obtained from the observational data. For instance, the vertical structure function is said to be optimized at the eddy center. Is the eddy center at x1 or x2? What if the location of eddy center changes with depth when an eddy tilts vertically?*

*Furthermore, is the vertical structure, when normalized by its maximum, consistent at different radial distances from the eddy center? In addition, Zhang et al. (2013) proposed a universal horizontal structure function for mesoscale eddies (their equation (2)), which includes a sign reversal of eddy pressure anomalies, beyond approximately 1.4 times the eddy radius (their figure 2). However, this feature does not appear to be present in figure 8 of the manuscript. I suggest comparing the observed horizontal structure to the function proposed by Zhang et al. (2013) to provide a better justification for the proposed framework.*

We thank the reviewer for this helpful comment. The first step of our optimization process is the decomposition of the observed anomaly into spiciness and heaving modes, as described in Section 3.4. Subsequently, the optimization is performed first for the vertical structure and then for the horizontal structure. In practice, however, the order is not critical because the variables are decoupled. We chose to optimize the vertical structure first to highlight the novelty of our approach, which lies in the formulation of the vertical structure function.

The vertical structure function is indeed optimized at the eddy center, defined as the location where the orthogonal velocity to the ship transect is zero. If the eddy center varies with depth, we use its average location. We have clarified this point in the revised manuscript (see line 374). In our dataset, the sampled eddies remain nearly vertical, with minimal variation in their center positions. As a result, we chose to keep $x_1$ and $x_2$ constant; however, it is possible to introduce a depth dependence, $x_1(z)$ and $x_2(z)$, to account for potential drift. Our formulas are intended as a flexible basis for eddy modeling and can be adapted to specific cases as needed.

Regarding the consistency of the vertical structure, , we confirm that it remains valid at different radial distances from the eddy center, as demonstrated in Figure 13. In this figure, we reconstructed the two-dimensional vertical structure of the sampled eddies using our model and compared it with in situ data. The agreement remains good at various radial distances, with a notable increase in error only beyond $R_1$.

We have also attempted to fit the observed horizontal structure with the function proposed by Zhang et al. (2013):

$$\phi(r) = (1 - (r/R_0)^2) \, exp(- (r/R_0)^2) \quad (1)$$

For comparison, our model uses:

$$\phi(r) = exp(- (r/R_1)^\alpha) \quad (2) \text{ (corresponding to Eq. 27 in our manuscript)}$$

The function (1) fits well the data when $\alpha = 2$, as shown in panels (a) and (d) in Figure 8. However, our data do not extend beyond $r/R_1 > 1.5$, so we cannot observe the change of sign of $\phi(r)$ as reported by Zhang et al. (2013; see their figure 2).

Moreover, function (1) does not fit our data well for $\alpha = 4 \, or \, 6$. We provide two illustrative examples below, where the blue curve represents the data and the orange curve is the best fit using the Zhang et al. (2013) model. The steepness of funtion (1) is insufficient to

represent the sharp boundary of the anticyclones sampled during EUREC4A-OA, making (2) a more appropriate choice.

Based on these results, and to avoid further lengthening an already extensive manuscript, we have decided not to include the Zhang et al. (2013) function in our analysis.

[Figure]

We have added a sentence in line 192-195 to clarify this point.

_3 Minor Comments_

_1. Equation (53) and (54): Is the α enforced to be an integer? It seems that if α is odd and x−x1 is negative, the eddy anomaly will increase exponentially from its center, which is unphysical. Perhaps, the absolute value should be taken for x−x1?_

We thank the reviewer for this careful observation. The absolute value is now correctly included in equations (53) and (54) in the revised version of the manuscript.

_2. I am not sure whether the tilde above all variables is necessary. It seems that it does not have a special meaning and is in fact dropped in later sections. To make the derivations clearer, I suggest dropping tilde completely._

We thank the reviewer for his/her comment. This suggestion was initially raised by a previous reviewer who found it difficult to distinguish between functions and values due to the abundance of notations. We agree that the change has improved the clarity of the article, especially given the complexity of the notation used.

_3. Line 119: I think η should be the vertical displacement of isopycnal with respect to the mean state, not the state of rest._

We thank the reviewer for this remark. We agree with the suggestion and have modified the expression accordingly in the revised manuscript (see line 120).

_4. Line 134: I suggest dropping "we assume". You can estimate the typical density variation corresponding to the isopycnal height change, not subjectively assuming it._

We thank the reviewer for his/her suggestion. The wording has been modified in the revised manuscript accordingly (see line 134).

_5. Lines 141 and 170: Missing spaces between ˜·and texts._

We thank the reviewer for bringing this to our attention. The typographic errors have been corrected in the revised manuscript (see lines 142 and 171).

_6. Equation (21): What's the difference between T(r,θ,σ0) and T(σ0)? It feels that equation (21) is just zero equals zero._

We thank the reviewer for this remark. $\widetilde{T}(r, \theta, \sigma_0)$ represents the in situ temperature field, while $\widetilde{\overline{T}}(\sigma_0)$ denotes the mean state. In the presence of an eddy with a spiciness mode anomaly, the temperature is not constant along isopycnal surfaces. For the same density, multiple (T,S) pairs can coexist. Therefore, equation (21) does not simply express "zero equals zero.".

*7. Line 185: I do not think it is accurate to say "no exact analytical expression for φ(r) exists". Zhang et al. (2013) has proposed an analytical expression for the radial structure of eddies.*

We thank the reviewer for bringing this to our attention. The reviewer is indeed correct,. and we have modified the sentence accordingly in the revised version (see line 186).

*8. Line 208: "radius" →"radial distance".*

We thank the reviewer for bringing this to our attention. The wording has been corrected in the revised manuscript (see line 219).

*9. Line 223: I suggest drop "assuming" for the same reason mentioned before.*

We thank the reviewer for this suggestion. We agree and have removed the word "assuming" in the revised manuscript (see line 234).

*10. Line 248: H1 is a characteristic depth scale.*

We thank the reviewer for bringing this to our attention. We agree with the suggestion, and the wording has been modified accordingly in the revised manuscript (see line 259).

*11. Table 1: A "vertical" is not capitalized.*

We thank the reviewer for bringing this to our attention. We agree, and the expression has been corrected in Table 1 of the revised manuscript.

*12. Line 268: How exactly is the eddy center be determined from velocity analysis?*

We thank the reviewer for his/her relevant comment. The sentence in question has been deleted from the revised manuscript, as the method for determining the eddy center is already described in Section 3.1.4 (lines 320-322). Specifically, we used the methodology described in Nencioli et al (2008) to estimate the position of the eddy center in the (x,y) plane.

We would also like to clarify that this eddy center is different from the eddy center identified along a ship cross-section, where it is defined as the location where the orthogonal velocity to the ship transect is zero. To clarify this point, we have added a sentence in lines 383–384 of the revised manuscript.

*13. Table 2: The filter scale for x is different for different observations. What's the justification for the filter scale? Is it based on the local deformation radius?*

We thank the reviewer for his/her question. The choice of thresholds (or cutoff scale) is somewhat subjective and depends on the spatial scales under investigation. Our primary aim was to minimize the influence of submesoscale effects. Therefore, we selected a cutoff value slightly larger than the mean resolution of the raw data.

*14. Line 319 and others: "cutoff period" →"cutoff scale" or "filter scale".*

We thank the reviewer for bringing this to our attention. This is indeed the "cutoff scale", and the expression has been corrected in the revised manuscript.

*15. Line 367: ψ0 should be ηz0.*

We thank the reviewer for bringing this to our attention. This notation was carried over from a previous version of the manuscript and is no longer relevant in the current. The expression has been corrected in the revised version (see line 380).

*16. Line 373: "ψ and ξ" →"ηz and δ2 z σ"? I see no ψ and ξ in equations (53)-(56).*

We thank the reviewer for bringing this to our attention. This notation originated from a previous version of the manuscript and is not relevant in the current context). The expression has been corrected in the revised version (see line 386).

*17. Line 374: There is an extra period.*

We thank the reviewer for bringing this to our attention. The typographic error has been corrected in the revised version (see line 387).

*18. Line 374: "locations where the amplitude of ψ and ξ are maximal" →"depth where the amplitude of ηz and δ2 z σ are maximal"?*

We thank the reviewer for bringing this to our attention. This notation originated from a previous version of the manuscript and is not relevant for the current version. The expression has been corrected in the revised version (see line 388).

*19. Line 389: Why is the water in an anticylconic eddy be colder than its surroundings?*

We thank the reviewer for his/her relevant question. The statement that anticyclonic eddies always transport warmer water than their surroundings is not universally valid. Both positive and negative temperature anomalies can occur, as demonstrated, for example, in the study of Aguedjou et al (2021).

*20. Figure 8: Which depth are the φ and χ from? The depth where ηz is the maximum? Is φ(x) or χ(x) different at other depth?*

We thank the reviewer for his/her relevant remark. In Figure 8, Φ and X are fitted at the depth where ηz is maximum.

Regarding the reviewer's second question, it was indeed important to demonstrate that the optimization works at all depths. This is the purpose of Figure 13, which shows the error between the in situ density field and the reconstructed field. The error remains small within the eddy cores, indicating that Φ and X are nearly  constant with depth.

*21. Line 441: There is an extra "function".*

We thank the reviewer for bringing this to our attention. The extraneous occurrence of "function" has been deleted in the revised manuscript (see line 455).

*22. Figures 9 and 10: Which location are the vertical profiles from? The eddy center? Is ηz (z) or δ2 z σ(z) different at different x?*

We thank the reviewer for his/her remark. Yes, the vertical profiles are from the eddy center. Here, the eddy center is the appearing eddy center refers to its position on the two-dimensional cross-section, defined as the location where the orthogonal velocity to the ship transect is zero.

As previously mentioned, it was indeed important to demonstrate whether the optimization was effective at different x-locations.. This is addressed in Figure 13, which shows the error between the in situ density field and the reconstructed field. We found that, $\eta_z(z)$ or $\delta_z^2(z)$ remain valid when $r$ is less than the radius of maximum velocity.

*23. Line 465: The theory here has more tuning parameters than the models of Flierl (1987) and Zhang et al. (2013). The comparison between them seems to be unfair.*

We thank the reviewer for his/her remark. We fully agree and discuss this point in the Discussion section (see lines 512 and 517).

*24. Line 508: "reconstruct" →"reconstructed".*

We thank the reviewer for bringing this to our attention. The error has been corrected in the revised manuscript.

*225. Figure 14: Labels a, b, and c are repeated in the second row, and there is no alphabetic label in the third row. Please correct them.*

We thank the reviewer for his/her remark. The assignment of one column to each letter, with each letter corresponding to a specific eddy, was an intentional choice. We agree that the third row was missing a letter, and we have corrected the figure accordingly in the revised version.

*26. Line 549: "diffusivity" →"diffusion".*

We thank the reviewer for bringing this to our attention. The error has been corrected in the revised manuscript.

*27. Line 553: There should be more details about how the vertical extent is controlled by the stratification and what the vertical symmetry means.*

We thank the reviewer for his/her comment. Additional details have been included in the conclusion of the revised manuscript (see lines 571 and 574).

**References**

*Chelton, D. B., M. G. Schlax, and R. M. Samelson, 2011: Global observations of nonlinear mesoscale eddies. Prog. Oceanogr., 91 (2), 167–216.*

*Flierl, G., 1987: Isolated eddy models in geophysics. Annu. Rev. Fluid Mech., 19 (1), 493–530.*

*Zhang, Z., Y. Zhang, W. Wang, and R. Huang, 2013: Universal structure of mesoscale eddies in the ocean. Geophys. Res. Lett., 40, 3677–3681.*

**References**

Aguedjou, H. M. A., Chaigneau, A., Dadou, I., Morel, Y., Pegliasco, C., Da‑Allada, C. Y., & Baloïtcha, E. (2021). What can we learn from observed temperature and salinity isopycnal anomalies at eddy generation sites? Application in the tropical Atlantic Ocean. Journal of Geophysical Research: Oceans, 126(11), e2021JC017630.

Nencioli, F., Kuwahara, V. S., Dickey, T. D., Rii, Y. M., & Bidigare, R. R. (2008). Physical dynamics and biological implications of a mesoscale eddy in the lee of Hawai'i: Cyclone Opal observations during E-Flux III. Deep Sea Research Part II: Topical Studies in Oceanography, 55(10-13), 1252-1274.

Zhang, Z., Y. Zhang, W. Wang, and R. Huang, 2013: Universal structure of mesoscale eddies in the ocean. Geophys. Res. Lett., 40, 3677–3681.

---

## Author Response (AR2)

**Reply referee 2**

I appreciate the authors' thorough response to my previous comments and the substantial revisions made. The manuscript is now clearer and better presented. I have just a few minor suggestions to help further refine the argument.

We would like to thank the reviewer again for his/her review and constructive feedback. We appreciate the effort and time the reviewer has invested in evaluating our work. Please find our point-to-point response below in blue.

Are  $x_1$  and  $x_2$  estimated by fitting or by the location of the vertically averaged center? Line 380 indicates that they are one of the parameters obtained by optimization/fitting, but from the authors' response to my comments, it seems that  $x_1$  and  $x_2$  are estimated as the location of vertically averaged eddy center on cross-sections. Please clarify this.

We thank the reviewer for his/her remark. We apologize if our previous response was not clear enough. The location of the vertically averaged eddy center is used to optimize the vertical structure function only as a starting step. This is consistent with what we wrote in the article between line 385 and 390. x\_1 and x\_2 are estimated by fitting. It explains the differences in values between x1 and x2.

Equation (9) and others: I found the notation \delta^2 to be confusing. It can also mean the square of \delta or the second-order derivative operator. To avoid this confusion, I suggest using a different symbol, such as \delta^s for spiciness mode.

We thank the reviewer for his/her suggestion. We have adopted the notation \delta^s in the revised manuscript.

Line 137: "assume" -> "estimate"

We thank the reviewer for his/her remark. The wording has been corrected in the revised version of the manuscript.

Equation (53) and (54): Is \alpha enforced to be an integer during the optimization? If so, please say that in the manuscript.

We thank the reviewer for bringing this to our attention. Yes, \alpha is enforced to be an integer. We have clarified this point in the revised version.

Equation (27) and (28): \alpha\_1 and \alpha\_2 have already been used for different variables in equation (22).

We thank the reviewer for bringing this to our attention. Notations have been modified in the revised version.

Line 387: vertically averaged location

We thank the reviewer for his/her suggestion. The wording has been modified in the revised version of the manuscript.

Figure 14: I still think it is uncommon to assign several panels with the same label. It would be clearer to assign them with six different letters such as (a)-(i).

We thank the reviewer for his/her suggestion. We have modified the figure accordingly in the revised version.

Caption of Figure 14: (Nencioli et al., 2008) -> the approach of Nencioli et al., (2008)

We thank the reviewer for bringing this to our attention. The sentence has been modified accordingly.

Line 277: Remove either "at" or "near".

We thank the reviewer for bringing this to our attention. The sentence has been modified accordingly.